# Recurrent Equivariant Constraint Modulation: Learning Per-Layer Symmetry Relaxation from Data

Stefanos Pertigkiozoglou [1]   Mircea Petrache [2]   Shubhendu Trivedi [3]   Kostas Daniilidis [1]

## Abstract

Equivariant neural networks exploit underlying task symmetries to improve generalization, but strict equivariance constraints can induce more complex optimization dynamics that can hinder learning. Prior work addresses these limitations by relaxing strict equivariance during training, but typically relies on prespecified, explicit, or implicit target levels of relaxation for each network layer, which are task-dependent and costly to tune. We propose Recurrent Equivariant Constraint Modulation (RECM), a layer-wise constraint modulation mechanism that learns appropriate relaxation levels solely from the training signal and the symmetry properties of each layer's input-target distribution, without requiring any prior knowledge about the task-dependent target relaxation level. We demonstrate that under the proposed RECM update, the relaxation level of each layer provably converges to a value upper-bounded by its symmetry gap, namely the degree to which its input-target distribution deviates from exact symmetry. Consequently, layers processing symmetric distributions recover full equivariance, while those with approximate symmetries retain sufficient flexibility to learn non-symmetric solutions when warranted by the data. Empirically, RECM outperforms prior methods across diverse exact and approximate equivariant tasks, including the challenging molecular conformer generation on the GEOM-Drugs dataset.

## 1. Introduction

Equivariant neural networks have emerged as a key paradigm for incorporating known task symmetries into machine learning models. By constraining network layers to respect the underlying task symmetry, these architectures achieve improved generalization and robustness across a large range of domains (Cohen & Welling, 2016; Kondor & Trivedi, 2018; Bekkers, 2020; Bronstein et al., 2021). Despite their successes, a growing body of work observes that in certain tasks, even when the underlying symmetries are present, replacing equivariant models with their unconstrained counterparts results in more stable training and improved performance (Wang et al., 2024; Abramson et al., 2024). Recent works conjecture that this phenomenon arises because imposed symmetry constraints in the parameter space may induce a more complex optimization landscape (Nordenfors et al., 2025), restricting available optimization trajectories and potentially trapping them in suboptimal regions of the parameter space (Xie & Smidt, 2025).

These observations have motivated efforts to relax equivariance constraints during training while imposing them back on the model at test time. Pertigkiozoglou et al. (2024) and Manolache et al. (2025) propose using approximate equivariant networks, originally developed for tasks with misspecified or approximate symmetries, as a way to improve training dynamics even when exact symmetries hold. Specifically, they demonstrate that relaxing the equivariant constraints during training and reimposing them at test can improve performance while retaining the parameter and sample efficiency of equivariant architectures.

While these solutions have been shown to generalize across various tasks and equivariant architectures, they lack a principled way to choose *where* to relax the equivariance constraint and by *how much*. This limitation makes their applicability to a new model nontrivial. The most effective choice of modules used for relaxing the equivariant constraint can differ between models and tasks. The optimal combination and the level of relaxation depends on both the architecture and the specific symmetries of the task, with no known principled way to choose a priori. The aforementioned methods partially address this challenge by providing additional knowledge about the level of relaxation the individual

[1]University of Pennsylvania [2]Pontificia Universidad Católica de Chile [3]Fermi National Accelerator Laboratory; Now at Google DeepMind. Correspondence to: Stefanos Pertigkiozoglou, Shubhendu Trivedi <pstefano@seas.upenn.edu, shubhendu@csail.mit.edu>.

*Proceedings of the 43rd International Conference on Machine Learning*, Seoul, South Korea. PMLR 306, 2026. Copyright 2026 by the author(s).

layers should achieve by the end of training. This information is encoded either through explicitly designed relaxation schedules (Pertigkiozoglou et al., 2024) or through additive penalty terms whose weighting implicitly determines the target level of relaxation (Manolache et al., 2025). This requirement is further complicated by the fact that optimal relaxation levels, as well as the most effective combination of modules used to relax equivariance, can differ across layers of the same model. Without additional prior knowledge, discovering the appropriate level of relaxation of each layer requires extensive hyperparameter tuning, making the approach computationally expensive and difficult to scale.

In this work, we address this limitation by designing a training framework that performs per-layer modulation of different relaxation techniques using only the symmetries present in the task supervision, without requiring prior knowledge of the specific level of relaxation required for the task. Specifically, we propose a relaxation layer and an update rule with the following properties:

1. Each layer is described as a linear combination of an equivariant component and weighted unconstrained components. Relaxation can be applied to any subset of network components, and their weights are recurrently updated during training using a learned update rule.

2. The update rule guarantees that the weights of each unconstrained component converge to a value upper bounded by the distance between the learned data distribution and its symmetrized counterpart. Thus, layers with fully symmetric input-target distributions converge to be equivariant without requiring any additional hyperparameter tuning, while layers with non-symmetric distributions retain the flexibility to relax equivariant constraints and learn approximate equivariant functions. *Here, the input-target distribution refers to the joint distribution over a layer's input and the model's ground truth target output used as supervision.*

This convergence guarantee is a key contribution distinguishing our approach from prior work: rather than requiring practitioners to specify a relaxation scheduler or additional optimization terms, our framework automatically discovers the appropriate level of equivariance for each layer based on the symmetry structure of its input-target data. This enables our method to modulate equivariant constraints across tasks with both approximate and exact symmetries, and architectures with different optimal relaxation strategies.

## 2. Related Work

The principle of incorporating symmetry directly into NN architectures has a long and heterogeneous intellectual lineage. Early instances include the work of Fukushima (1979;

1980) inspired by neurophysiological studies of visual receptive fields (Hubel & Wiesel, 1962; 1977), culminating in the CNN architectures formalized by LeCun et al. (1989). In parallel, *Perceptrons* (Minsky & Papert, 1987) articulated a more general program for NNs grounded in group invariance theorems, a line of inquiry that was actively pursued through the mid-1990s (Shawe-Taylor, 1989; Wood & Shawe-Taylor, 1993; Shawe-Taylor, 1993; 1994). This program was revived with a generalized CNN-based and representation-theoretic perspective in Cohen & Welling (2016), with subsequent extensions to continuous and higher-dimensional symmetry groups (Weiler et al., 2018a;b). A unified prescriptive mathematical theory of such networks was developed by Kondor & Trivedi (2018); Cohen et al. (2019); Weiler et al. (2024). These ideas have been applied to different domains (Esteves et al., 2018; Maron et al., 2019), and have been successful in a wide range of applications including 3D vision (Deng et al., 2021; Chatzipantazis et al., 2023), molecular modeling (Jumper et al., 2021; Hoogeboom et al., 2022; Batzner et al., 2022), language modeling (Petrache & Trivedi, 2024; Gordon et al., 2020), and robotics (Zhu et al., 2022; Ordoñez-Apraez et al., 2024).

Despite their extensive successes, a major limitation of equivariant neural networks is the assumption of exact distributional symmetries. As argued theoretically by Petrache & Trivedi (2023), misspecifying the level of symmetry in a task can degrade generalization. To avoid such a misspecification Zhou et al. (2021) proposes learning the appropriate task symmetries and equivariant weight constraints through a meta-learning optimization loop. Another popular response has been the design of approximate equivariant architectures that interpolate between strict equivariance and fully unconstrained networks. Finzi et al. (2021) proposed adding an unconstrained component parallel to the exact equivariant layers, while Wang et al. (2022) relaxed weight sharing in group convolutional or steerable networks by allowing small perturbations between otherwise shared parameters. Romero & Lohit (2022) introduced methods for learning partial equivariance, and Gruver et al. (2023) developed approaches for measuring learned equivariance. Additionally, van der Ouderaa et al. (2023) optimized for the appropriate level of relaxation through a Bayesian model selection framework, requiring specialized KFAC (Martens & Grosse, 2015) approximations to make the model's Hessian computation tractable. More recently, Veefkind & Cesa (2024) proposed projection back to the equivariant parameter space to control relaxation, while Berndt & Stühmer (2026) used a similar projection as an equivariance-promoting regularizer. Ashman et al. (2024) designed an architecture-agnostic approximate equivariant framework applied to neural processes.

While the above approximate equivariant models improve performance in tasks lacking exact symmetries, they do not

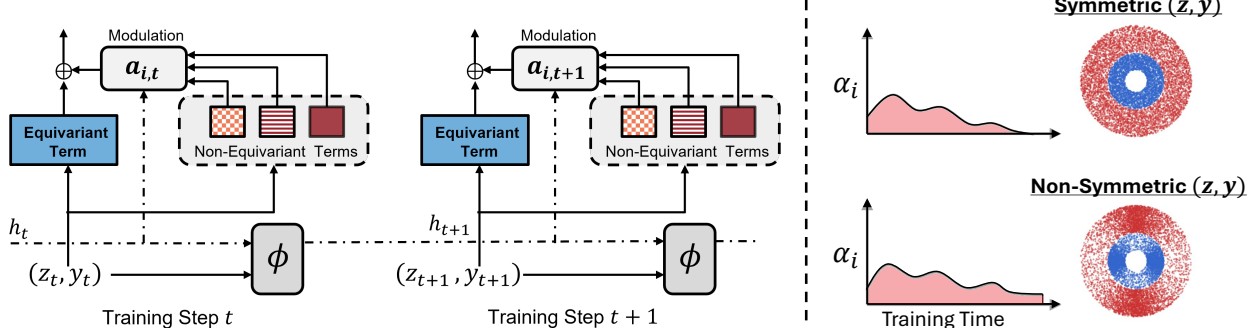

*Figure 1.* [Left] In our proposed Recurrent Equivariant Constraint Modulation (RECM) framework, the equivariant layer is relaxed by the addition of non-equivariant terms modulated by $\alpha_{i,t}$. These modulation parameters are controlled by an optimization state vector $h_t$, which is recurrently updated at each training step through the update rule described in Equation 1 (visualized as $\phi$ here).[Right] RECM guarantees that in cases of fully symmetric input-target distributions the non-equivariant modulation parameters $\alpha_i$ converge to zero, recovering fully equivariant layers, while in non-symmetric cases the network has flexibility to learn approximate equivariant solutions.

explain recent empirical observations showing that unconstrained models can outperform equivariant models even when exact symmetries are present (Wang et al., 2024; Abramson et al., 2024). Nordenfors et al. (2025) contrasted optimization trajectories of equivariant and augmentation-based approaches, highlighting possible limitations of exactly equivariant neural networks, while Xie & Smidt (2025) showed that equivariance can obscure parameter-space symmetries and fragment the optimization landscape into disconnected basins. Elesedy & Zaidi (2021) sketched a projected gradient method for constructing equivariant networks, suggesting that relaxation during optimization could be beneficial, but without empirical validation.

More recently, Pertigkiozoglou et al. (2024) aimed to address some of these limitations by proposing a scheduling framework to modulate equivariant constraints during training with projection back to the equivariant parameter space at test time. Building on this work, Manolache et al. (2025) formulated the problem as a constrained optimization, enabling adaptive constraint modulation via dual optimization without ad hoc scheduling. While this approach learns constraint modulation dynamically, it still requires an implicitly predefined target level of equivariance for each layer. Our proposed framework builds on these works and aims at addressing one of their limitations: the need for prior knowledge of the desired equivariance level in the final trained model.

## 3. Preliminaries

Before presenting our proposed method, it is useful to clearly define the equivariant constraints imposed on each individual layer and their relaxation. Given a group $G$ we can define its action on a vector space $V$ through a linear representation i. e. a map $\rho : G \to GL(V)$ which satisfies the group homomorphism property $\rho(g_1)\rho(g_2) = \rho(g_1 g_2)$

for any two elements $g_1, g_2 \in G$. This linear representation allows us to map each element $g \in G$ to an invertible linear map $\rho(g) \in GL(V)$ that acts on a vector $v \in V$. A layer of a neural network $f : \mathbb{R}^n \to \mathbb{R}^k$ is equivariant to a group $G$ acting to its input and output by representations $\rho_{\text{in}}(g), \rho_{\text{out}}(g)$ if for every $v \in \mathbb{R}^n$:

$$f^{\text{eq}}(\rho_{\text{in}}(g)v) = \rho_{\text{out}}(g)f^{\text{eq}}(v), \quad \forall g \in G$$

When $f^{\text{eq}}$ is a linear map parametrized by a matrix $W_{\text{eq}}$, it must satisfy the constraint $W_{\text{eq}}\rho_{\text{in}}(g) = \rho_{\text{out}}(g)W_{\text{eq}}$ for all $g \in G$.

An extensive body of work now exists to solve this constraint, characterizing the space of equivariant matrices (or intertwiners) for various groups $G$ and input-output representations pairs (Weiler et al., 2018a; Thomas et al., 2018; Deng et al., 2021; Maron et al., 2019). Following Finzi et al. (2021), we can relax the equivariant constraint of this linear layer by creating a convex combination of an equivariant linear layer and an unconstrained affine layer $f(x) = \beta W_{\text{eq}}x + \alpha_1 W_{\text{un}}x + \alpha_2 b$.

## 4. Method

While the relaxation presented in Section 3 can be effective when we have prior knowledge of the correct weighted combination of equivariant and non-equivariant terms for each layer, recovering an effective combination by only using the task supervision can be non-trivial. To address this, we first propose to consider a weighted sum of multiple "candidate" non-equivariant terms, and design an update rule that recovers their modulation weights such that they optimize for the task performance while respecting its underlying symmetries. Specifically, we define each layer as:

$$f_{\theta,\alpha,\beta} = \beta f_{\theta_0}^{\text{eq}} + \sum_{i=1}^{K} \alpha_i f_{\theta_i}^{\text{un}_i}$$

and recurrently update $\beta$, $\alpha_i$ during training until convergence to the weighted combination that is used during inference. This is an extension of the setting used in Pertigkio-zoglou et al. (2024) and Manolache et al. (2025), where a single unconstrained term was considered per layer.

In order to recover $\alpha_i, \beta$, Pertigkiozoglou et al. (2024) set $\beta = 1$ and use a linear scheduler for $\alpha_i$ that starts from 0, is linearly increased to 1, and then set to 0 at the end of training. On the other hand, Manolache et al. (2025) formulate the problem as a constrained optimization: $\alpha_i$ is optimized along with the model's parameters by solving the dual problem. In both cases, the users must know apriori the useful level of symmetry constraints and impose them by either designing a scheduler for $\alpha_i$ or by formulating the appropriate constrained optimization problem. As discussed in Section 1, while the general symmetry of a task's data distribution is often accessible, its interaction with intermediate model features is complex—governed by model architecture, task specifics, and optimization dynamics rather than by simple known rules. Additionally, in both of the above cases the modulation of the relaxation constraints can alter the overall optimization objective. This is clear in the case of Pertigkiozoglou et al. (2024), where the modulation parameters are changing independently of the task loss gradient. In the case of Manolache et al. (2025), the dual problem can correspond to an altered version of the original objective if the required relaxation level is misaligned with the task, creating a trade-off between task loss and constraint satisfaction.

The above observations motivate our approach: allowing the model to learn per-layer requisite symmetries by learning an update rule that enforces the model's equivariance to scale with the symmetry of the input-target distribution. More concretely, as discussed in Section 1, we require the parameter $\alpha_i$ of each layer to converge to a value whose modulus is upper bounded by some measure of the invariance gap of its input-target distribution, and it satisfies the following:

1. In cases of fully invariant input-target distributions, $\alpha_i$ should converge to zero, and thus we recover an equivariant layer.
2. For non-invariant distributions the network should be free to learn unconstrained non-invariant solutions. This ensures we do not over-constrain the network when the data lacks the desired symmetry.

### 4.1. Recurrent Equivariant Constraint Modulation

To formalize both requirements, we consider a model $f$ composed of $N$ learnable layers $f^{(j)}$ and $N$ optional standard parameter-free layers $\sigma^{(j)}$ (e.g. non-learnable nonlinearities, skip connections, or identity layers):

$$f = \sigma^{(N)} \circ f^{(N)} \circ \sigma^{(N-1)} \circ f^{(N-1)} \circ \ldots \sigma^{(1)} \circ f^{(1)}$$

We denote the intermediate input representations $z_t^{(j)}$ at layer $j$ at optimization step $t$, defined recursively as:

$$z_t^{(1)} := x_t,$$
$$z_t^{(j+1)} := \sigma^{(j)}(f^{(j)}(z_t^{(j)})), \quad j = 1, \ldots, N$$

Additionally, we can associate the intermediate representation $z_t^{(j)}$ with the ground truth target output $y_t$ that is used in the loss $L(f(x_t), y_t)$ at optimization step $t$. As a result for each layer $j$ at optimization step $t$ we create pair $(z_t^{(j)}, y_t)$.

The goal of our proposed method is to simultaneously learn the parameters of each learnable component $f^{(j)}$ along with their appropriate level of constraint modulation. To achieve this, we parametrize each layer $f^{(j)}$ to use dynamic and layer specific modulation parameters $\beta_t^{(j)}, \alpha_{i,t}^{(j)}$. Since we aim for a unified treatment of all layers we design a parametrization and an update rule that is applied independently to each $f^{(j)}$ using their input-target pairs $(z_t^{(j)}, y_t)$ at each optimization step $t$. The parametrization and update rule take identical forms across layers (differing only in their layer-specific state variables and learnable parameters); thus, to simplify notation, we present them without the superscripts $(j)$ for the remainder of this work. First, each layer is parametrized as:

$$f_{\theta, \alpha_t, \beta_t}(z_t) := \beta_t f_{\theta_{\text{eq}}}^{\text{eq}}(z_t) + \sum_{i=1}^{K} \alpha_{i,t} f_{\theta_{\text{un},i}}^{\text{un},i}(z_t)$$

with the modulation parameters $\alpha_{i,t}, \beta_t$ being controlled by an optimization state variable $h_t$ and are updated as follows:

$$h_t = \left(1 - \frac{a}{b + at}\right) h_{t-1} + \frac{a}{b + at} l_{\theta_t}(z_{t-1}, y_{t-1}) \quad (1)$$

$$\alpha_{i,t} = s(w_{\alpha_i}^T h_t), \quad \beta_t = k(w_\beta^T h_t), \quad (2)$$

where $h_t$ is a optimization state vector (separate for each layer), $s, k$ are point-wise non-linear functions with $k(0) = 1, s(0) = 0, l_{\theta_t} : \mathbb{R}^{n_{\text{in}}} \times \mathbb{R}^{n_{\text{out}}} \to \mathbb{R}^m$ is a learnable update rule, $w_{\alpha_i}, w_\beta \in \mathbb{R}^m$ are learnable vectors with bounded norm $\|w_{\alpha_i}\| \leq 1, \|w_\beta\| \leq 1$, and $a, b$ are scalars that control the decay speed of the exponential weighted average.

As optimization progresses, each layer updates all the parameters of equivariant layers, unconstrained layers, and the update rule $(l_\theta, w_{\alpha_i}, \beta)$ simultaneously through gradient descent, while the update of the optimization state $h_t$ is performed by Equation 1. This formulation uses an exponential weighted average for the update of $h_t$ with time-varying weights that provide flexibility to the learning algorithm to dynamically change the equivariant constraint, while also allowing us to quantify and control the convergence of $h_t$. In practice, for an optimization with $T$ total iterations, we can set $b = 1$ and adjust $a$ accordingly so that at $T$ steps,

$a/(b + aT)$ reaches a value close to zero. Here, it is important to note that $a$ must be tuned for a specific model and training setup once, and it can then be used to train the same model on different data distributions with different symmetry properties. This distinguishes our method from previous work that requires the regularization terms controlling the level of relaxation to be tuned for each individual training distribution. Another important property of the above update rule is that, for intermediate layers, the distribution of the pairs $(z, y)$ changes dynamically, since the input $z$ is a learnable feature output of a previous layer. Nevertheless, given a reasonable assumption about the convergence of learnable parameters, the following results provide guarantees about the convergence of $h_t$.

**Lemma 4.1** (Convergence of $h_t$). **[Proof provided in Appendix A]** *Assume at time $t$ we sample $(z_t, y_t)$ from a distribution with density $p_t$, where $p_t$ converges in 1-Wasserstein distance to distribution $p$. Also assume that $l_{\theta_t}$ is bounded, L-Lipschitz and converges uniformly to $l^*$. Then we have that*

$$h_t \underset{a.s.}{\to} \mathbb{E}_{(z,y) \sim p}[l^*(z, y)] = h^*.$$

Here, it is important to note that the assumptions of convergence of both the distribution $p_t$ and the update function $l_\theta$ can be easily satisfied by using a learning rate scheduler that has the learning rate converging to zero at the end of training. Since such learning rate schedulers are commonly used in practice, with the most popular example being the cosine annealing scheduler (Loshchilov & Hutter, 2017), the assumptions on convergence of Lemma 4.1 can be easily satisfied in most training frameworks. Figure 2, along with Appendix B, showcases different cases of the convergence of parameters $\alpha_i$ for both symmetric and non-symmetric distributions. Using the result of Lemma 4.1, we can control the convergence of $h_t$ by bounding the expectation of the limit state $h^* = \mathbb{E}_{(z,y) \sim p}[l^*(z, y)]$. The recurrent state update proposed in the Recurrent Equivariant Constraint Modulation (RECM) framework, along with its convergence properties, is illustrated in Figure 1. In the next section, we present an efficient design of an update function $l_\theta$, such that $h^*$ satisfies the required properties presented at the beginning of this section.

### 4.2. Design of update function $l_\theta$

For the limit state $h^*$, the required properties are:

- The absolute value of each element of $h^*$ is upper bounded by the distance between the input-target distribution $p$ (distribution at convergence) and its invariant projection $p_G := \int_G p(\rho_{in}(g)z, \rho_{out}(g)y)dg$. This property guarantees that in the cases where $p$ is invariant $h^* = 0$ and since $s(w_{\alpha,i}^T 0) = 0$ and $k(w_\beta^T 0) = 1$, we recover a fully equivariant layer.

| | Rotated | | Aligned | |
|---|---|---|---|---|
| Model | Inst. | Cls. | Inst. | Cls. |
| VN-PointNet | 0.68 | 0.62 | 0.68 | 0.62 |
| VN-PN+ES. | 0.72 | 0.67 | 0.67 | 0.62 |
| VN-PN+RPP | 0.77 | 0.71 | 0.89 | **0.86** |
| VN-PN+ELLA | 0.78 | 0.71 | 0.90 | **0.86** |
| VN-PN+RECM | **0.80** | **0.74** | 0.90 | **0.86** |

*Table 1.* Classification Accuracy on ModelNet40 (Chang et al., 2015) dataset using a baseline VN-PointNet network and different versions of constraint relaxation, including our proposed RECM

- When $p$ is not invariant, the recurrent update is free to converge to $h^*$ with elements not equal to zero and thus the learning dynamics can converge to non-equivariant solutions. So when $p(z, y) \neq p(\rho_{in}(g)z, \rho_{out}(g)y)$ for some $g \in G$ and $(x, y) \in Z$, there exists $\theta^*$ such that $h^* = \mathbb{E}_{(z,y) \sim p}[l_{\theta^*}(z, y)] \neq 0$.

Before defining the update function, we first need to define a generating set for the group of interest $G$. Specifically, if $G$ is a topological group we say that $C_G \subset G$ is a **topological generating set** (or simply generating set) of $G$ if the topological closure of $C_G^\star := \{w = g_1 g_2 \ldots g_n : n \in \mathbb{N}, g_1, \ldots, g_n \in C_G\}$ is the whole $G$. A probability measure is said to be **adapted** if it has support equal to a generating set $C_G$. For example, if $G = SO(3)$, a topological generating set is $C_{SO(3)} = \{R_1, R_2\}$ where $R_1$ is the rotation by $\pi/2$ around the $x$-axis and $R_2$ is the rotation by $\pi/3$ around the $z$-axis (for the proof, see Appendix Lemma A.1).

Given the above, for a given group $G$ and a topological generating set $C_G$ we can define the update function $l_\theta$ as:

$$l_\theta(z, y) = r_\theta(z, y) - \frac{1}{|C_G|} \sum_{g \in C_G} r_\theta(\rho_{in}(g)z, \rho_{out}(g)y),$$
(3)

with $r_\theta$ being parametrized by an MLP, or by any other parametrization that ensures Lipschitz continuity with a bounded constant. Since an MLP satisfies the Lipschitz assumption (Virmaux & Scaman, 2018) we can expect convergence of $h_t$ as shown in Lemma 4.1. In order to be able to provide the guarantees on the expectation of $l_\theta$, described at the beginning of this section, we focus on compact groups where a normalized Haar measure over the group exists. This setting covers a broad range of groups of interest, including the rotation groups $SO(3)$ and $SO(2)$, their finite subgroups (e.g., cyclic and dihedral groups), and finite groups such as the permutation group $S_n$. Additionally, while the hidden state $h_t$ is a multi-dimensional vector, because the update rule is applied "pointwise" and independently at each dimension, we provide results of convergence for scalar $h_t$. The results can be trivially extended to the

*Table 2.* Comparison of our proposed framework RECM with prior work on exact equivariant and approximate equivariant tasks. Mean squared error achieved by different equivariant models and method leveraging constraint relaxation applied on: (**left**) the equivariant task of N-body simulation (Kipf et al., 2018), (right) the task of motion capture trajectory prediction (CMU, 2003).

**N-body Simulation**

| Method | MSE ↓ |
|---|---|
| SE(3)-Tr. | 24.4 |
| RF | 10.4 |
| EGNN | 7.1 |
| EGNO | 5.4 |
| SEGNN | $5.6_{\pm 0.25}$ |
| SEGNN$_{ES}$ | $4.9_{\pm 0.18}$ |
| SEGNN$_{RPP}$ | $4.1_{\pm 0.17}$ |
| SEGNN$_{ELLA}$ | $3.9_{\pm 0.11}$ |
| SEGNN$_{ACE\text{-}exact}$ | $3.8_{\pm 0.16}$ |
| SEGNN$_{ACE\text{-}appr}$ | $3.8_{\pm 0.17}$ |
| SEGNN$_{RECM}$ | $\mathbf{3.7}_{\pm 0.22}$ |

**Motion Capture Trajectory Prediction**

| Model | MSE↓ (Run) | MSE↓ (Walk) |
|---|---|---|
| EF | $521.3_{\pm 2.3}$ | $188.0_{\pm 1.9}$ |
| TFN | $56.6_{\pm 1.7}$ | $32.0_{\pm 1.8}$ |
| SE(3)-Tr. | $61.2_{\pm 2.3}$ | $31.5_{\pm 2.1}$ |
| EGNN | $50.9_{\pm 0.9}$ | $28.7_{\pm 1.6}$ |
| EGNO | $33.9_{\pm 1.7}$ | $8.1_{\pm 1.6}$ |
| EGNO$_{ES}$ | $33.1_{\pm 1.2}$ | $7.8_{\pm 0.3}$ |
| EGNO$_{RPP}$ | $28.0_{\pm 1.6}$ | $7.2_{\pm 0.2}$ |
| EGNO$_{ELLA}$ | $29.9_{\pm 1.1}$ | $7.0_{\pm 0.4}$ |
| EGNO$_{ACE-exact}$ | $32.6_{\pm 1.6}$ | $7.5_{\pm 0.3}$ |
| EGNO$_{ACE-appr}$ | $23.8_{\pm 1.5}$ | $7.4_{\pm 0.2}$ |
| EGNO$_{RECM}$ | $\mathbf{22.6}_{\pm 0.8}$ | $\mathbf{6.6}_{\pm 0.5}$ |

vector case by applying them independently for each component (see Appendix A.1):

**Theorem 4.2.** *[Proof provided in Appendix A] Let $p$ be a probability distribution on metric space $Q = Z \times Y$, let $G$ be a compact group with Haar measure $\lambda_G$ acting on $Q$ by continuous unitary representations $\rho_{in}, \rho_{out}$ i.e. $T_g(z, y) = (\rho_{in}(g)z, \rho_{out}(g)y)$, under which measure $p$ is preserved and let $C_G \subset G$ be a finite topological generating set of $G$. Define:*

$$p_{C_G} = \frac{1}{|C_G|} \sum_{g \in C_G} p(T_g^{-1}z), \quad p_G = \int_G p(T_g^{-1}z)d\lambda_G(g).$$

*Then if $l_\theta$ defined in (3) with $r_\theta : Q \to \mathbb{R}$ is a B-Lipschitz function, we have that*

$$\left| \mathbb{E}_{(z,y)\sim p}[l_\theta(z,y)] \right| \leq 2BW_1(p, p_G),$$

*where $W_1$ is the 1-Wasserstein distance between probability measures. Additionally, if $\{r_\theta\}_{\theta \in \Theta}$ is a family of universal approximators of all bounded continuous functions with Lipschitz constant less than or equal to a $C > 0$, then for any $C' \in (0, C)$ there exists $\theta^* \in \Theta$ for which $\mathbb{E}_{(z,y)\sim p}[l_{\theta^*}(z,y)] \geq C'W_1(p, p_{C_G})$.*

Using the proposed $l_\theta$, we can guarantee that its expectation, and thus the converging absolute values of $h_t$ are upper bounded by the distance between the input-target distribution and its symmetrized equivalent. By implementing the $s$ non-linearity of Equation 2, using a GeLU (Hendrycks & Gimpel, 2016), and given that $\|w_{a_i}\| \leq 1$, the relaxation modulation $a_{i,t}$ converges to $|a_i^*| = |s(w_{a_i}^T h^*)| \leq 2BW_1(p, p_G)$ for the case of scalar $h_t$ and to $|a_i^*| \leq 2\sqrt{m}BW_1(p, p_G)$ for the general case of a $m$-dimensional state vector $h_t$ (see Appendix A.1 for details). This overall bound thus verifies the first desired property of RECM.

For the second property to hold, we need to show that given an expressive enough function approximator $r_\theta$ (e.g. an MLP (Hornik et al., 1989)), there exist parameters $\theta^*$ for which $h_t$ does not converge to zero for non-symmetric distributions. Theorem 4.2 shows that there exists $\theta^*$ such that $h^* = \mathbb{E}_{(z,y)\sim p}[l_{\theta^*}(z,y)] > C'W_1(p, p_{C_G})$, thus the second property is equivalent of showing that for any non-symmetric distribution $p$ we have that $W_1(p, p_{C_G}) > 0$ meaning $p \neq p_{C_G}$. This is a consequence of the following:

**Lemma 4.3.** *[Proof provided in Appendix A] Let $p$ be a probability measure over $Q = Z \times Y$, let $C_G$ be a finite topologically generating set of the compact group $G$. For an action $T_g(z, y) = (\rho_{in}(g)z, \rho_{out}(g)y)$ by continuous representations $\rho_{in}, \rho_{out}$ on $Q = Z \times Y$, define $p_{C_G} := \frac{1}{|C_G|} \sum_{g \in C_G} p \circ T_g^{-1}$. Then the following holds*

$$p = p_{C_G} \iff p = p \circ T_g \quad \forall g \in G.$$

Theorem 4.2 and Lemma 4.3 show that our proposed update rule and function $l_\theta$ allow the model to freely learn the level of modulation for the equivariant and non-equivariant distributions, using only the task supervision, while guaranteeing that in cases where the distribution of intermediate feature and ground truth outputs is fully invariant the corresponding layer will converge to an equivariant solution.

## 5. Experiments

### 5.1. Ablation Study

We first verify the effectiveness of our proposed recurrent constraint modulation by performing ablation studies on the task of shape classification. Specifically, we use the lightweight VN-PointNet architecture (Deng et al., 2021) to classify the categories of sparsely sampled point clouds

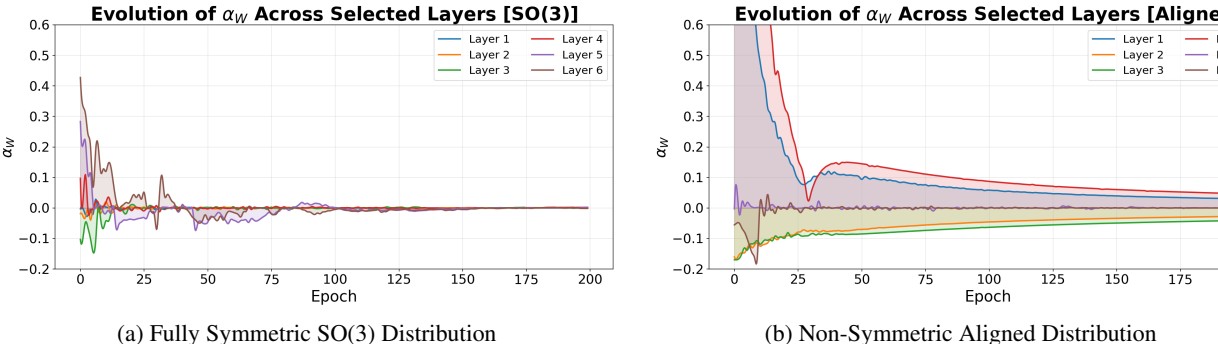

(a) Fully Symmetric SO(3) Distribution

(b) Non-Symmetric Aligned Distribution

*Figure 2.* Evolution of relaxation modulation parameter $\alpha_w$ of RECM applied on ModelNet (Chang et al., 2015) classification of a full SO(3) symmetric dataset (input shapes with random poses) and a non-symmetric aligned dataset ( input shapes with pre-aligned poses)

(300 points) from the ModelNet40 (Chang et al., 2015) dataset. This experiment studies the behavior of the update layer proposed in 4.1 on target distributions with different types of symmetry. Thus, we evaluate our modified model (VN-PointNet+RECM) on a "Rotated" dataset where pointclouds are rotated by a random SO(3) rotation and a "Aligned" dataset that uses the ModelNet40 aligned pointclouds. For the relaxation of the equivariant constraints, we follow the formulation presented in Section 3 with the addition of an additive noise term with a learnable standard deviation parameter $\sigma^2$. As a result, we replace all linear layers of VN-PointNet with their relaxed counterparts with form $f(x) = \beta_t W_{eq} x + \alpha_{W,t} W_{un} x + \alpha_{b,t} b + \alpha_{noise,t} \epsilon_s$ with $\epsilon \sim \mathcal{N}(0, \sigma I)$ being sampled Gaussian noise. Since modulation parameters $\alpha$ converge close enough to zero, but not at exactly zero, we remove the additive non-equivariant terms from contributing to the output of the layer if $|\alpha| < 0.01$. (See Appendix C for more experimental details)

Table 1 shows the final test instance and class accuracy achieved by a baseline VN-PointNet, a model where we schedule the relaxation of the equivariant constraint using a fixed schedule similar to (Pertigkiozoglou et al., 2024) (VN-PointNet+ES.), a model that we leave fully unconstrained to learn the added relaxation term (VN-PointNet+RPP) (Finzi et al., 2021), a model where the level of relaxation is chosen through the Bayesian model selection framework proposed in van der Ouderaa et al. (2023) (VN-PointNet+ELLA) and a model with our proposed recurrent equivariant modulation update (VN-PointNet+RECM). We observe how our proposed recurrent modulation adjusts to the symmetries of the task distribution and outperform all baselines in both the rotated and the aligned version without any additional adjustment required in any of the two cases.

Figure 2 illustrates how the parameters $\alpha_{W,t}$ of different layers are updated by RECM during training on the aligned and rotated dataset. Appendix B also provides the curves for the remaining relaxation terms $\alpha_{b,t}, \alpha_{noise,t}$. In the "Rotated" case, modulation for the unconstrained terms converges to

*Table 3.* Molecule conformer generation coverage recall and precision on GEOM-DRUGS ($\delta = 0.75$Å). RECM is applied to two different equivariant models ETFlow and DiTMC and compared to prior equivariant and non-equivariant models.

|  | Recall ↑ | | Precision ↑ | |
|---|---|---|---|---|
|  | mean | median | mean | median |
| GeoDiff | 42.10 | 37.80 | 24.90 | 14.50 |
| GeoMol | 44.60 | 41.40 | 43.00 | 36.40 |
| Torsional Diff. | 72.70 | 80.00 | 55.20 | 56.90 |
| MCF | 79.4 | 87.5 | 57.4 | 57.6 |
| ETFlow-Eq | 79.53 | 84.57 | 74.38 | 81.04 |
| ETFlow+RECM | 79.44 | 85.64 | 75.10 | 82.02 |
| DiTMC-Eq | 80.8 | 85.6 | 75.3 | 82.0 |
| DiTMC-Unc | 79.9 | 85.4 | 76.5 | 83.6 |
| DiTMC+RECM | 80.6 | 85.5 | 76.1 | 83.1 |

zero, with different layers showcasing different convergence rates. This result verifies the theoretical results of Section 4.2 about the convergence to a fully equivariant model in symmetric distributions without the need for pre-specifying per-layer levels of relaxation. On the other hand, in the "Aligned" case, the state convergence captures the lack of symmetry, with some layers converging to non-equivariant solutions. After removing the inactive layers with small $\alpha$, the "Rotated" trained model keeps 2M active parameters while the model trained on aligned distribution keeps 4.5M active parameters, showcasing how RECM dynamically adjusts the active parameters of the model used during inference based on the symmetries of the underlying distributions.

## 5.2. Comparisons On Tasks Requiring Different Level of Relaxation

In this section we showcase the performance of RECM in different tasks requiring different level of equivariant constraints. We apply our proposed framework both to the *fully equivariant* task of N-body simulation (Kipf et al., 2018), and the task of motion capture trajectory prediction (CMU,

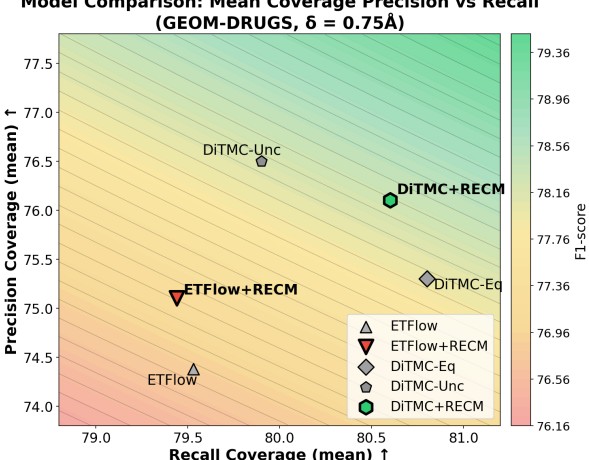

*Figure 3.* Coverage Precision and Recall achieved on GEOM-DRUGS ($\delta = 0.75$Å) for different method along with there combined F1 score. The F1 score of each point of the two dimensional diagram is visualized by the underlying color, with green regions corresponding to points with higher F1-score values. Additionally, the diagonal lines correspond to points with the same F1 scores.

2003), which, as discussed in Manolache et al. (2025), benefits more from approximate equivariant constraints. To evaluate RECM we compare with prior works that learn the level of relaxation only through the task loss (RPP) (Finzi et al., 2021), utilize a predefined schedule of constraint modulation (Pertigkiozoglou et al., 2024) projecting back to the equivariant parameter space by the end of training (ES), select the level of relaxation through Bayesian model selection (ELLA) (van der Ouderaa et al., 2023), or implicitly impose and optimize for the exact (ACE-exact) or the approximate equivariant constraint (ACE-appr) (Manolache et al., 2025). While both tasks are equivariant to roto-translations, RECM requires a compact group (to admit a normalized Haar measure). We therefore relax only the constraint controlling the rotation equivariance, leaving translation equivariance intact.

**(N-body Simulation)** The N-body simulation task consists of predicting the positions of 5 electrically charged particles after 1000 timesteps, given their initial positions, charges, and velocities. This task models the charges as free-moving particles without any external force, and thus it is fully equivariant to $SE(3)$ roto-translations. As the baseline on top of which we apply RECM, we utilize the SEGNN model proposed by (Brandstetter et al., 2021). In Table 2 we compare the error achieved by different methods that utilize constraint relaxation along with equivariant baselines such as the Equivariant Flows (EF) (Köhler et al., 2019), the SE(3)-Transformer (Fuchs et al., 2020) and SEGNN. In this simple, fully equivariant task, RECM outperforms all other baselines, including prior work that leverages constraint relaxation on top of SEGNN and fully equivariant models.

**(Motion Capture)** While the N-body simulation was a synthetic task designed to be fully equivariant, we are interested in investigating the ability of our framework to adjust its level of relaxation in tasks where exact symmetry is not optimal. Specifically, we consider the task of trajectory prediction in motion capture, where the goal is to predict future trajectories from input motion capture sequences. As shown in Manolache et al. (2025), this task benefits from equivariant architectures but achieves optimal performance with approximate equivariant networks. We use the setup proposed by Xu et al. (2024), which includes the base version of EGNO. We compare with both equivariant baselines such as Equivariant Flows (EF), Tensor Field Networks (TFN) (Thomas et al., 2018), SE(3)-Transformer (SE(3)-Tr.), EGNN and EGNO (Xu et al., 2024) as well the equivariant EGNO using the previously proposed constraint modulation approaches (Pertigkiozoglou et al., 2024; Manolache et al., 2025) EGNO$_{ES}$, EGNO$_{ACE-exact}$ and EGNO$_{ACE-appr}$. As shown in Table 2, while previous works are required to train both the exact equivariant and approximate equivariant architectures to conclude the optimal level of relaxation, our method can recover the appropriate relaxation parameters and achieve the best performance with a single training run.

In both tasks, RECM is able to adjust the expressivity of the models by modulating the constraint relaxation so that it matches the required task symmetry. In Appendix C.1, we provide a more detailed presentation of the active parameters at inference using RECM compared to the baseline models.

### 5.3. Conformer Generation

To demonstrate the scalability of our proposed framework, we apply it on the large scale GEOM-Drugs dataset (Axelrod & Gomez-Bombarelli, 2022), to solve the task of molecular conformer generation. In this task, the goal of the model is to generate the low-energy 3D structures (local energy minima) given only their molecular graph structure as input. An active debate is ongoing within the machine learning community, regarding whether or not equivariant architectures are actually necessary for achieving optimal performance for this task. While Jing et al. (2022) (Torsional Diff) showed significant benefits of incorporating an equivariant network, a later work by Wang et al. (2024) (MCF) described a fully unconstrained model achieving state-of-the-art results. More recent works investigate reintroducing implicit equivariance bias by structurally constraining the generation model (Hassan et al., 2024; Frank et al., 2025) (ETFlow,DiTMC), showing improvements in the generation precision. This ongoing debate makes the conformer generation task ideal for applying RECM, since, contrary to the previous methods, it does not require a predefined target level of equivariant constraint satisfaction.

Following the experimental setup of Hassan et al. (2024),

we evaluate our method when applied on the equivariant ETFlow and on the equivariant DiTMC-Eq, which is a diffusion transformer with equivariant positional encoding. We compare with previous equivariant methods GeoDiff (Xu et al., 2022), GeoMol (Ganea et al., 2021), Torsional Diff and the non-equivariant MCF. Additionally, for DiTMC, we compare with both its equivariant variant DiTMC-Eq discussed above and its non-equivariant variant DiTMC-Unc, which replaces the equivariant linear layers with unconstrained ones and uses non-equivariant relative positional encoding. We refer the reader to Appendix C.2 for a detailed description of the evaluation metrics.

In Table 3 we show the mean and median coverage precision and recall of the different methods. While the fully unconstrained MCF achieves state-of-the-art recall, it has a very low precision. On the other hand, several equivariant methods achieve significantly improved precision while still maintaining competitive recall. When applying our RECM framework, we observe that we can further improve the precision of the generation of the equivariant models while retaining the competitive recall. This is more apparent in Figure 3, where we plot together the precision and recall of the different versions of ETFlow and DiTMC overlayed over the F1-score (their harmonic mean). In both cases, applying RECM improves the overall F1-score compared to both the equivariant and the unconstrained variants, which supports the main argument of this work about the benefits of equivariant constraints adjusted to the specific task. Finally in Appendix C.3, in addition to the task performance, we show the appropriate level of relaxation recovered by RECM for the task of conformer generation.

## 6. Conclusion

In this work, we introduced Recurrent Equivariant Constraint Modulation (RECM), a framework designed to allow a model to adapt the level of relaxation of its equivariant constraint without requiring any prior designed relaxation schedules or equivariant-enforcing penalties. We provided theoretical guarantees demonstrating how RECM correctly converges to equivariant models in the case of a fully symmetric distribution, while it provides the flexibility for models to converge to approximate equivariant solutions in cases of non-symmetric tasks. Empirical evaluations across different equivariant and non-equivariant tasks validate the theoretical results and demonstrate that RECM consistently outperforms existing baselines. Our code is available at https://github.com/StefanosPert/Recurrent_Constraint_Modulation.

## Acknowledgements

SP and KD thank NSF FRR 2220868, NSF IIS-RI 2212433, ONR N00014-22-1-2677 for support. MP thanks the support of National Center for Artificial Intelligence CENIA, ANID Basal Center FB210017. ST was supported by the Computational Science and AI Directorate (CSAID), Fermilab. This work was produced by Fermi Forward Discovery Group, LLC under Contract No. 89243024CSC000002 with the U.S. Department of Energy, Office of Science, Office of High Energy Physics. The United States Government retains, and the publisher, by accepting the work for publication, acknowledges that the United States Government retains a non-exclusive, paid-up, irrevocable, world-wide license to publish or reproduce the published form of this work, or allow others to do so, for United States Government purposes. The Department of Energy will provide public access to these results of federally sponsored research in accordance with the DOE Public Access Plan (http://energy.gov/downloads/doe-public-access-plan).

## Impact Statement

This paper presents work whose goal is to advance the practice of equivariant neural networks within machine learning. As such, the potential societal consequences of our work are indirect and aligned with the general progress in the field, none of which we feel must be specifically highlighted here.

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

# A. Proof of main results

**Lemma 4.1 (Convergence of $h_t$)** Assume $(z_t, y_t)$ are independent random samples from a distribution with density $p_t$, where $p_t$ converges in 1-Wasserstein distance to $p$. Also assume that $l_{\theta_t}$ is bounded, $L$-Lipschitz and converges uniformly to $l^*$. Then:

$$h_t \underset{a.s.}{\to} E_{(z,y)\sim p}[l^*(z, y)].$$

*Proof.* We can define the random variable $Y_t = l_{\theta_t}(z_t, y_t)$, and its average over timesteps $k \le t$ as $\bar{Y}_t = \frac{1}{t}\sum_{k=1}^t Y_k$. Using the recursion we get:

$$
\begin{aligned}
h_t &= \frac{b + a(t-1)}{b + at}h_{t-1} + \frac{a}{b + at}Y_t \\
&= \frac{b}{b + at}h_0 + \frac{at}{b + at}\left(\frac{1}{t}\sum_{k=1}^t Y_k\right) \\
&= \lambda_t h_0 + (1 - \lambda_t)\bar{Y}_t,
\end{aligned}
$$

with $\lambda_t = \frac{b}{b+at}$. Now define $Z_t = l^*(z_t, y_t)$, then we have that:

$$\left|\frac{1}{t}\sum_{k=1}^t (Y_k - Z_k)\right| \le \frac{1}{t}\sum_{k=1}^t |Y_k - Z_k|,$$

and the above quantity tends to zero a.s. as $t \to \infty$, since $|Y_k - Z_k| \le \|l_{\theta_k} - l^*\|_\infty$, which is assumed to tend to zero.

Additionally define $\mu_k := \mathbb{E}_{(z,y)\sim p_k}[l^*(z, y)]$ and $\mu := \mathbb{E}_{(z,y)\sim p}[l^*(z, y)]$. Since $l^*$ is Lipschitz with Lipschitz constant $L$, we have that:

$$
\begin{aligned}
\left|\mathbb{E}_{(z,y)\sim p_k}[l^*/L] - \mathbb{E}_{(z,y)\sim p}[l^*/L]\right| &\le W_1(p, p_k) \implies \\
|\mu_k - \mu| &\le LW_1(p, p_k) \implies \\
\left|\frac{1}{t}\sum_{k=1}^t \mu_k - \mu\right| &\underset{t\to\infty}{\to} 0 \quad \left(\text{since } \lim_{t\to\infty}\frac{1}{t}\sum_{k=1}^t W_1(p_k, p) = \lim_{t\to\infty} W_1(p_t, p) = 0\right).
\end{aligned}
$$

Finally, for the random variables $(Z_k - \mu_k)$ for different $k \ge 1$, we know that they are independent with zero mean. Additionally, since $l$ is bounded we have that $\|l^*\|_\infty \le M$, and thus the absolute value of $Z_k$ and $\mu_k$ is also bounded by $M$. This means that the variance of $(Z_k - \mu_k)$ is:

$$\mathrm{Var}(Z_k - \mu_k) \le \mathbb{E}_{p_k}\left[(Z_k - \mu_k)^2\right] \le 4M^2.$$

Since $\mathbb{E}_{p_k}(Z_k - \mu_k) = 0$ for all $k$ and $\sum_{k=1}^\infty \mathrm{Var}(Z_k - \mu_k)/k^2 < \infty$ we have that:

$$\frac{1}{t}\sum_{k=1}^t (Z_k - \mu_k) \underset{t\to\infty}{\to} 0,$$

which implies:

$$\frac{1}{t}\sum_{k=1}^t Z_k - \mu = \frac{1}{t}\sum_{k=1}^t (Z_k - \mu_k) + \frac{1}{t}\sum_{k=1}^t (\mu_k - \mu) \underset{t\to\infty}{\to} 0 \implies \frac{1}{t}\sum_{k=1}^t Z_k \underset{t\to\infty}{\to} \mu = \mathbb{E}_{(z,y)\sim p}[l^*(z, y)].$$

Combining all of the above, we have:

$$\bar{Y}_t = \frac{1}{t}\sum_{k=1}^t (Y_k - Z_k) + \frac{1}{t}\sum_{k=1}^t Z_k \underset{t\to\infty}{\to} \mathbb{E}_{(z,y)\sim p}[l^*(z, y)].$$

With this and the fact that $\lim_{t\to\infty}\lambda_t = \lim_{t\to\infty}\frac{b}{b+at} = 0$ allow us to finalize the proof:

$$h_t = \lambda_t h_0 + (1 - \lambda_t)\bar{Y}_t \underset{t\to\infty}{\to} 0h_0 + 1\mathbb{E}_{(z,y)\sim p}[l^*(z, y)] = \mathbb{E}_{(z,y)\sim p}[l^*(z, y)].$$

$\square$

**Theorem 4.2** Let $p$ be a probability distribution on metric space $Q = Z \times Y$, let $G$ be a compact group with Haar measure $\lambda_G$ acting on $Q$ by continuous unitary representations $\rho_{\text{in}}, \rho_{\text{out}}$ i.e. $T_g(z, y) = (\rho_{\text{in}}(g)z, \rho_{\text{out}}(g)y)$, under which measure $p$ is preserved and let $C_G \subset G$ be a finite topological generating set of $G$. Define:

$$p_{C_G} = \frac{1}{|C_G|} \sum_{g \in C_G} p(T_g^{-1}z), \quad p_G = \int_G p(T_g^{-1}z)d\lambda_G(g).$$

Then if $l_\theta$ defined in (3) with $r_\theta : Q \to \mathbb{R}$ a $B$-Lipschitz function, we have that

$$\left| \mathbb{E}_{(z,y)\sim p}[l_\theta(z, y)] \right| \leq 2B W_1(p, p_G),$$

where $W_1$ is the 1-Wasserstein distance between probability measures. Additionally, if $\{r_\theta\}_{\theta \in \Theta}$ is a family of universal approximators of all bounded continuous functions with Lipschitz constant less than or equal to a $C > 0$, then for any $C' \in (0, C)$ there exists $\theta^* \in \Theta$ for which $\mathbb{E}_{(z,y)\sim p}[l_{\theta^*}(z, y)] \geq C' W_1(p, p_{C_G})$.

*Proof.* For the first part of the proof, since $\rho_{\text{in}}, \rho_{\text{out}}$ are unitary representations and $r_\theta$ is a $B$-Lipschitz function we have from the definition of $l_\theta$ that:

$$\|l_\theta\|_{\text{Lip}} \leq \|r_\theta\|_{\text{Lip}} + \frac{1}{|C_G|} \sum_{g \in C_G} \|r_\theta \circ T_g\|_{\text{Lip}} = 2B.$$

Also the expected value of $r_\theta$ over the symmetrized distribution is:

$$\mathbb{E}_{q\sim p_G}[l_\theta(q)] = \mathbb{E}_{q\sim p_G}[r_\theta(q)] - \frac{1}{|C_G|} \sum_{g \in C_G} \mathbb{E}_{q\sim p_G}[r_\theta(T_g q)] = 0.$$

Since the function $f = \frac{1}{(2B)}l_\theta$ has Lipschitz constant less or equal to 1, from the Kantorovich-Rubinstein duality for the 1-Wasserstein distance we have that:

$$\begin{aligned}
|\mathbb{E}_{q\sim p}[l_\theta(q)]| &= 2B \left| \mathbb{E}_{q\sim p}\left[\frac{l_\theta(q)}{2B}\right] - \mathbb{E}_{q\sim p_G}\left[\frac{l_\theta(q)}{2B}\right] \right| \\
&\leq 2B \sup_{\|f\|_{\text{Lip}}} (E_{q\sim p}[f] - E_{q\sim p_G}[f]) \\
&\leq 2B\, W_1(p, p_G).
\end{aligned}$$

For the second part of the proof we compute:

$$\mathbb{E}_{q\sim p}[l_\theta(q)] = \mathbb{E}_{q\sim p}[r_\theta(q)] - \frac{1}{|C_G|} \sum_{g \in C_G} \mathbb{E}_{q\sim p}[r_\theta(T_g q)],$$

where we can write the second term as:

$$\begin{aligned}
\frac{1}{|C_G|} \sum_{g \in C_G} \mathbb{E}_{q\sim p}[r_\theta(T_g q)] &= \frac{1}{|C_G|} \sum_{g \in C_G} \int r_\theta(T_g q)p(q)dq \\
&= \frac{1}{|C_G|} \sum_{g \in C_G} \int r_\theta(u)p(T_{g^{-1}})du \\
&= \int r_\theta(u) \left( \frac{1}{|C_G|} \sum_{g \in C_G} p\left(T_{g^{-1}}u\right) \right) du \\
&= E_{q\sim p_{C_G}}[r_\theta(q)],
\end{aligned}$$

where we used the change of variable $u = T_g q$ and the fact that the group action preserves measure $p$. As a result, the overall expectation can be written as

$$\mathbb{E}_{q \sim p}[l_\theta(q)] = \mathbb{E}_{q \sim p}[r_\theta(q)] - \mathbb{E}_{q \sim p_{C_G}}[r_\theta(q)].$$

If $W_1(p, p_{C_G}) = 0$ then the second statement of the theorem follows from the first one, so we pass to the case $W_1(p, p_{C_G}) > 0$.

Similarly to before we have that by Kantorovich duality:

$$\sup_{\|f\|_{\mathrm{Lip}} \leq 1} \left( \mathbb{E}_{q \sim p}[f(q)] - \mathbb{E}_{q \sim p_{C_G}}[f(q)] \right) = W_1(p, p_{C_G})$$
$$\implies \forall \epsilon_1 > 0, \exists f : \|f\|_{\mathrm{Lip}} \leq 1 \text{ and } \mathbb{E}_{q \sim p}[f(q)] - \mathbb{E}_{q \sim p_{C_G}}[f(q)] \geq W_1(p, p_{C_G}) - \epsilon_1.$$

As $\{r_\theta\}_{\theta \in \Theta}$ are universal approximators of functions with Lipschitz constant less or equal to $C$, for every $f$ with $\|f\|_{\mathrm{Lip}} \leq 1$ and for all $\epsilon_2 > 0$ there exists $\theta^*$ such that $\|f - C^{-1} r_{\theta^*}\|_\infty < \epsilon_2/2$. Which implies that there exists $\theta^*$ such that:

$$\mathbb{E}_{q \sim p}[C^{-1} r_{\theta^*}(q)] - \mathbb{E}_{q \sim p_{C_G}}[C^{-1} r_{\theta^*}(q)] \geq W_1(p, p_{C_G}) - (\epsilon_1 + \epsilon_2) \implies \mathbb{E}_{q \sim p}[l_{\theta^*}(q)] \geq C W_1(p, p_{C_G}) - C(\epsilon_1 + \epsilon_2).$$

For $(\epsilon_1 + \epsilon_2) < \frac{C - C'}{C} W_1(p, p_{C_G})$ it follows that $\mathbb{E}_{q \sim p}[l_{\theta^*}(q)] > C' W_1(p, p_{C_G})$, as desired. $\square$

**Lemma 4.3** Let $p$ be a probability measure over $Q = Z \times Y$, let $C_G$ be a finite topologically generating set of compact group $G$. For an action $T_g(z, y) = (\rho_{in}(g)z, \rho_{out}(g)y)$ by continuous representations $\rho_{in}, \rho_{out}$ on $Q = Z \times Y$, define $p_{C_G} := \frac{1}{|C_G|} \sum_{g \in C_G} p \circ T_g^{-1}$. Then the following holds

$$p = p_{C_G} \iff p = p \circ T_g \text{ for all } g \in G.$$

*Proof.* For the backward direction ($\Leftarrow$) it suffices to note that:

$$\forall g \in C_G, \ p = p \circ T_{g^{-1}} \implies p_{C_G} = \frac{1}{|C_G|} \sum_{g \in C_G} p \circ T_g^{-1} = p.$$

For the forward direction ($\implies$) we define the uniform probability measure on the finite set $C_G \subset G$:

$$\mu = \frac{1}{|C_G|} \sum_{g \in C_G} \delta_g$$

and also define the convolution on a probability measure $\sigma$ on G as:

$$(p * \sigma)(q) = \int_G p(T_{g^{-1}} q) d\sigma(g).$$

Then we have

$$(p * \mu)(q) = \frac{1}{|C_G|} \sum_{g \in C_G} p(T_{g^{-1}} q) = p_{C_G}(q).$$

Applying the convolution with $\mu$ multiple times in the above result, we get that:

$$p = p_{C_G} = p * \mu \implies p = p * \mu^{*n}, \quad \forall n \geq 1$$

Then setting

$$\bar{\mu}_n := \frac{1}{n} \sum_{k=0}^{n-1} \mu^{*k},$$

we have that:

$$p * \overline{\mu}_n = \frac{1}{n} \sum_{k=0}^{n-1} p * \mu^{*k} = \frac{1}{n} \sum_{k=0}^{n-1} p = p.$$

Now since the support of $\mu$ generates a dense subgroup of the compact group $G$ ($\mu$ is adapted), we can use the Kawada-Itô theorem (Kawada & Itô, 1940; Grenander, 1963) to conclude that $\overline{\mu}_n$ converges weakly to the Haar measure $\lambda_G$ on G:

$$\overline{\mu}_n \overset{n \to \infty}{\Longrightarrow} \lambda_G.$$

This weak convergence implies that for every bounded continuous function $F : G \to \mathbb{R}$:

$$\int_G F(g) d\overline{\mu}_n(g) \longrightarrow \int_G F(g) d\lambda_G(g),$$

so for every bounded continuous function $f : Q \to \mathbb{R}$ we can define function:

$$F(g) := \int_Q f(q) p(T_{g^{-1}} q) dq,$$

which, since $f$ is bounded and $p$ is a probability distribution, it is also bounded, and because the action $T_g$ is continuous, it is also continuous. Using the above $F$ we have

$$\int_G F(g) d\overline{\mu}_n(g) = \int_G \int_Q f(q) p(T_{g^{-1}} q) dq d\overline{\mu}_n(g)$$
$$= \int_Q f(q) \int_G p(T_{g^{-1}} q) d\overline{\mu}_n(g) dq$$
$$= \int_Q f(q)(p * \overline{\mu}_n)(q) dq$$

and similarly

$$\int_G F(g) d\lambda_G(g) = \int_Q f(q)(p * \lambda_G)(q) dq,$$

which means that using the weak convergence of $\overline{\mu}_n$ we can get

$$\overline{\mu}_n \overset{n \to \infty}{\Longrightarrow} \lambda_G \implies \int_Q f(q)(p * \overline{\mu}_n)(q) dq \overset{n \to \infty}{\Longrightarrow} \int_Q f(q)(p * \lambda_G)(q) dq$$
$$\implies p * \overline{\mu}_n \overset{n \to \infty}{\Longrightarrow} p * \lambda_G.$$

Now, since for every $n \geq 1$, $p = p * \overline{\mu}_n$ we have that $p = p * \lambda_G$. Then for any $g \in G$, using the Dirac mass $\delta_{g^{-1}}$ at $g^{-1} \in G$, we get

$$p \circ T_g = p * \delta_{g^{-1}} = (p * \lambda_G) * \delta_{g^{-1}} = p * (\lambda_G * \delta_{g^{-1}}) = p * \lambda_G = p,$$

which conclude the proof for the forward direction. $\square$

**Lemma A.1.** *The set $C_{SO(3)} := \{R_1, R_2\}$ is a topological generating set for $G = SO(3)$ if $R_1, R_2$ are rotations that do not commute, do not generate a finite group, and do not jointly preserve any unoriented axis. This is the case e.g. in the following cases:*

1. *$R_1, R_2$ are rotations around distinct axes, that have infinite order (i.e. have irrational angles).*

2. *$R_1, R_2$ are the algebraic elements given below:*

$$R_1 = \begin{pmatrix} 1 & 0 & 0 \\ 0 & 0 & -1 \\ 0 & 1 & 0 \end{pmatrix}, \quad R_2 = \begin{pmatrix} 1/2 & -\sqrt{3}/2 & 0 \\ \sqrt{3}/2 & 1/2 & 0 \\ 0 & 0 & 1 \end{pmatrix}.$$

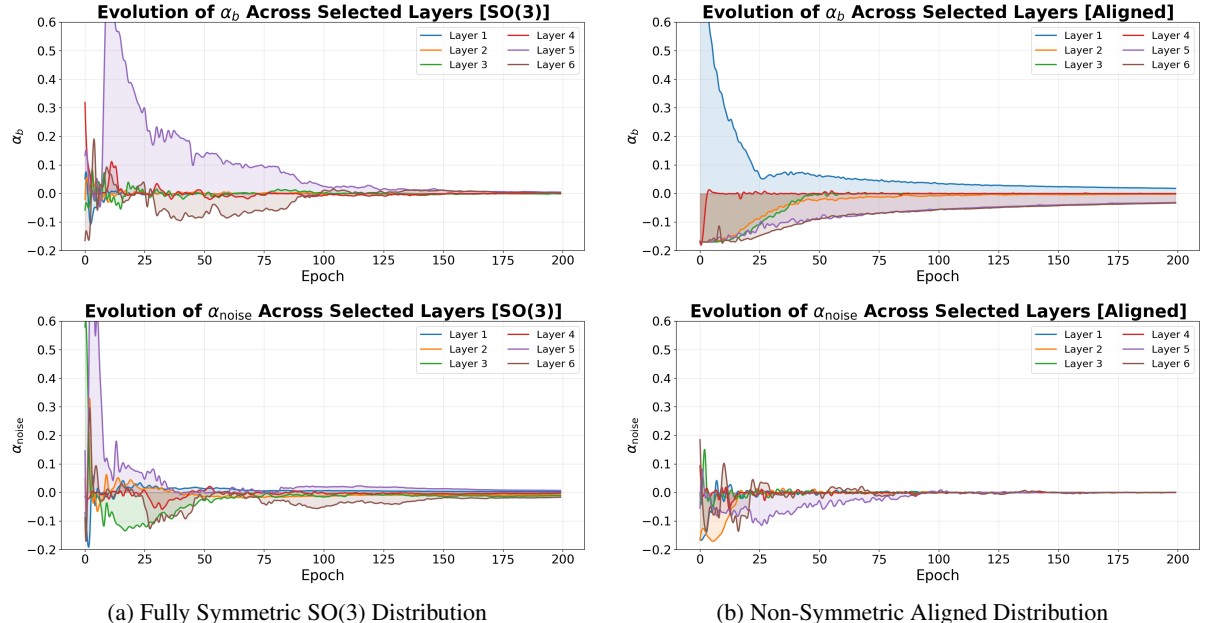

(a) Fully Symmetric SO(3) Distribution  (b) Non-Symmetric Aligned Distribution

*Figure 4.* Evolution of relaxation modulation parameter $\alpha_b, \alpha_{\text{noise}}$ of RECM applied on ModelNet40 (Chang et al., 2015) classification of a fully $SO(3)$ symmetric dataset (where the input shapes have random poses) and a non-symmetric aligned dataset (where the input shapes have pre-aligned poses)

*Proof.* We refer to the classification of closed subgroups of $SO(3)$, which are either finite, isomorphic to $SO(2)$ (in particular abelian), isomorphic to $O(2)$ (the stabilizer of an unoriented axis), or equal to $SO(3)$. If the group generated by $R_1, R_2$ is not one of the finite groups of $SO(3)$ (cyclic, dihedral, tetrahedral, octahedral, icosahedral), is not abelian (as is $SO(2)$), and is not contained in any $O(2)$, then its closure must be $SO(3)$ as desired.

As to the examples given, we note that $R_1, R_2$ being rotations around distinct axes which are not half turns, they do not commute, covering the first requirement. Further, in the first example the generated group is not finite because the two rotations have infinite order. In the second example, we can verify directly that the two axes and angles of rotation of $R_1, R_2$ are not compatible with any of the finite subgroups of $SO(3)$. Finally, in the first example $\{R_1, R_2\}$ is not contained in any $O(2)$: in $O(2) \subset SO(3)$ the only elements of infinite order are rotations around the preserved unoriented axis $\ell$, so containment in $O(2)$ would force both $R_1$ and $R_2$ to be rotations around $\ell$, contradicting the assumption that their axes are distinct. Similarly for the second example: since neither $R_1$ nor $R_2$ is a half-turn, joint preservation of an unoriented axis $l$ would force both to be rotations about $l$, which is a contradiction since their axes are $x$ and $z$ respectively. $\qquad\square$

### A.1. Extension to multivariable state $h_t$

In Section 4.2 we provided results for the simplified case of scalar state $h_t$. In the more general case of multidimensional $h_t$, since we apply the update rule of Equation 1 "pointwise" we can easily extend the provided result to consider a multidimensional state vector. Specifically consider $h_t \in \mathbb{R}^m$, where we can apply Theorem 4.1 and Theorem 4.2 at each dimension of $h_t$ independently to show that the $i^{\text{th}}$ element of $h_t$, which we denote as $h_{t,i}$, converges to $h_i^*$ with $|h_i^*| \leq 2BW_1(p, p_G)$.

Then given that $\|w_{\alpha_i}\| \leq 1$, we can compute the bound:

$$|w_{\alpha_i}^T h^*| \leq \|h^*\| \leq 2\sqrt{m}BW_1(p, p_G)$$

and given the nonlinearity $s$ implemented as a GeLU, with property $|s(x)| \leq |x|$, we have that overall $|s(w_{\alpha_i} h^*)| \leq 2\sqrt{m}BW_1(p, p_G)$.

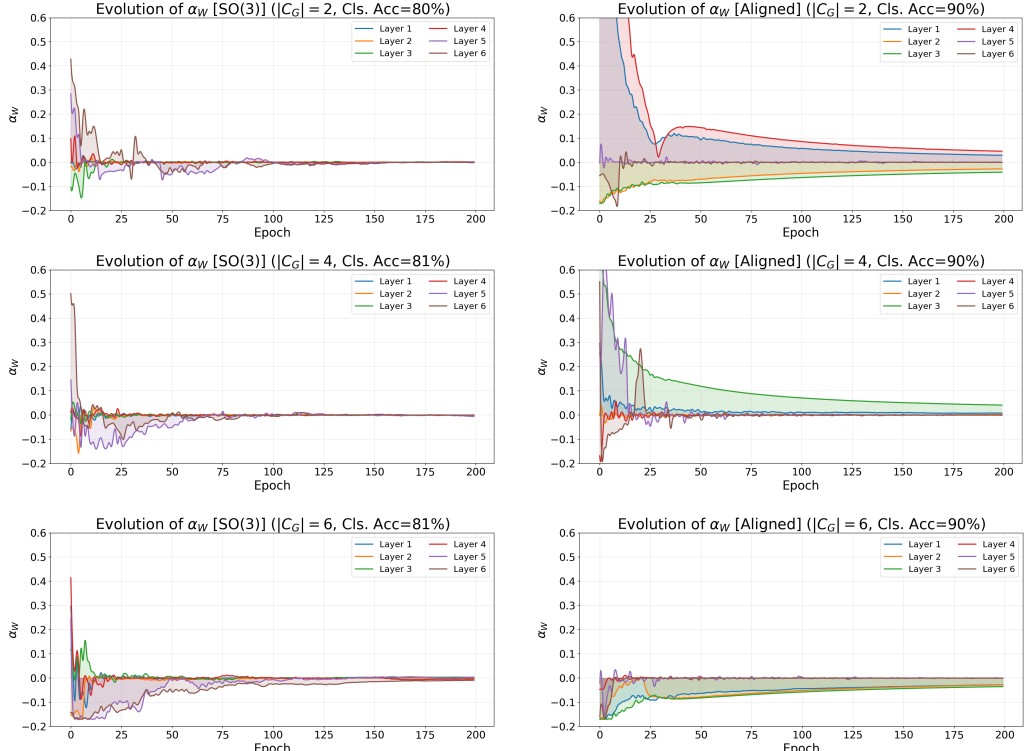

*Figure 5.* Evolution of relaxation modulation parameter $\alpha_W$ for different sizes of generating set $C_G$, when applying RECM on the VN-PointNet model trained on the ModelNet40 classification with a full symmetric SO(3) dataset and an aligned dataset.

## B. Modulation parameters evolution

In addition to Figure 2, Figure 4 shows the evolution of relaxation parameters $\alpha_{b,t}, \alpha_{\text{noise},t}$ for the ablation study on point cloud classification presented in Section 5.1. We can observe that the modulation parameter $\alpha_b$ for the additional bias term has a similar behavior as $\alpha_w$, namely converging to zero for the case of the SO(3) symmetric distribution, while converging to non-zero values for the case of the aligned distribution. On the contrary, the additional noise modulation parameter $\alpha_{\text{noise}}$ converges to zero in both cases, since reducing the output variance in both types of distributions improves performance on the deterministic classification task.

## C. Implementation Details

In this section, we provide the implementation details for the experimental evaluations presented in Section 5. For all experiments, we implement the optimization state variable $h_t$ using a 16 dimensional vector and the learnable update function $r_\theta$ described in Equation 3 using a two-layer MLP with a hidden dimension equal to 16 and GeLU nonlinearities. During the experimental evaluation, we observed that an MLP of size 16 was sufficiently expressive to provide the convergence properties shown in Figures 2, 4. Table 4 also shows how the different sizes of $m$ affect the overall model performance, where we can observe that the performance is relatively robust around $m = 16$, with $m = 32$ not providing any additional benefits.

For the update rule of Equation 1, we set $b = 1$, while $a$ is a task specific hyperparameter that depends on the total iterations each model is required for convergence. We tune $a$ by performing a simple grid search for different powers of 10, ranging from $10^{-1}$ to $10^{-5}$. Additionally, since $\alpha_i$ will converge to exact zero only at infinite time steps, we approximate the exact convergence by clipping all $|\alpha_i| < \epsilon$ to be equal to zero, for some small $\epsilon = 0.01$. Finally, following the setup of Manolache et al. (2025), since the norm of the added non-equivariant terms can dominate the constraint modulation, growing in scale faster than the constraint is reduced, we set an upper bound on their norms, guaranteeing that small values of $\alpha_i$ correspond to small non-equivariant contributions.

*Table 4.* Ablation results when training VN-PointNet+RECM on the rotated ModelNet40 dataset. **First Row:** Pointcloud classification instance accuracy when we implement the RECM update rule using MLPs with different sizes of hidden dimension m. **Second Row:** Average standard deviation of the relaxation parameter $a_w$ during RECM training with different batch sizes.

| **MLP hidden dimension** $m$ | 4 | 8 | 16 | 32 | |
|---|---|---|---|---|---|
| Inst. Accuracy | 0.78 | 0.80 | 0.80 | 0.79 | |
| **Training Batch size** | 12 | 18 | 24 | 30 | 36 |
| $\alpha_w$-std | 4.25 | 4.47 | 3.92 | 4.01 | 3.68 |

In all experiments as topological generating set of SO(3) we used:

$$R_1 = \begin{pmatrix} 1 & 0 & 0 \\ 0 & \cos(3/2) & -\sin(3/2) \\ 0 & \sin(3/2) & \cos(3/2) \end{pmatrix}, \quad R_2 = \begin{pmatrix} \cos(3/2) & -\sin(3/2) & 0 \\ \sin(3/2) & \cos(3/2) & 0 \\ 0 & 0 & 1 \end{pmatrix}$$

The choice of the above generating set of just two rotations allows us to limit the required forward passes through $r_\theta$. As shown in A.1, any pair of rotations with distinct axes and irrational angles would suffice. The above theoretical result is also verified by Figure 5, which shows both the evolution of $\alpha_W$ and the final achieved accuracy when training on the rotated and aligned point cloud classification dataset with different sizes of generating set $|C_G|$. We observe that the evolution of $\alpha_W$, although different across training runs, exhibits similar convergence properties regardless of the size of the generating set used. Additionally the different size of $C_G$ doesn't affect the final accuracy achieved by the RECM models.

Table 4 investigates how the batch size affects the variance of the relaxation parameters during training with RECM. For each layer, we compute the standard deviation of $a_w$ using a sliding window, and we report the average standard deviation across all windows and layers. We train VN-PointNet+RECM models with batch sizes $24s$ for different scaling factors $s$, with learning rate scaled by $s$, and we compute the standard deviation over proportional scaled windows of size $40/s$. We can observe that the overall variance remains stable, given the appropriate scale adjustments.

For the task specific hyperaparameters:

- For the ablation studies on the point cloud classification, we follow the training setup used in Pertigkiozoglou et al. (2024) and classify sparse point clouds (300 points) from the ModelNet40 dataset. In all ablations, we used $a = 0.001$.

- For the N-body simulation problem, we used the training setup used in Kipf et al. (2018), by setting $a = 0.001$.

- For the Motion Capture sequence prediction task, we follow the training and evaluation setup used in Xu et al. (2024) with $a = 10^{-4}$ for the run dataset and $a = 10^{-5}$ for the walk dataset.

- For the conformer generation we used the corresponding training setups described in Hassan et al. (2024) and Frank et al. (2025) with $a = 10^{-4}$.

### C.1. Discussion on Parameter Overhead and Computational Overhead

As discussed in the previous section, terms with small $\alpha_i$ values can be pruned from the network at inference time. Table 5 reports the percentage of additional parameters retained in the relaxed models relative to the equivariant baseline after pruning. We observe a clear distinction based on data symmetry: for datasets with symmetric distributions (ModelNet40 Rotated and N-body simulation), RECM recovers fully equivariant solutions, introducing no additional parameters at inference. In contrast, for tasks that can benefit from breaking exact equivariance (ModelNet40 Aligned, motion capture, and conformer generation), the network converges to partially non-equivariant solutions, retaining a portion of the relaxation parameters. RECM thus adaptively exploits symmetry breaking structure when present in the data.

Although pruning eliminates overhead at inference for symmetric tasks, the additional parameters still incur a cost during training. However, because the relaxation terms can be computed in parallel with the equivariant base layers, the total time overhead remains modest. While the total parameter count can increase by over 50%, training time increases by only approximately 40%. For challenging tasks where the optimal level of relaxation is unknown a priori, this modest additional training cost is typically preferable to exhaustively tuning per-layer relaxation levels across multiple training runs.

*Table 5.* Overhead ratios introduced by RECM relative to baseline models. *Parameter overhead* reports the ratio of additional parameters retained at inference time (after pruning terms with small $\alpha_i$) relative to the baseline parameter count. *Training time overhead* reports the ratio of increased training time relative to the baseline.

| Dataset | Baseline Model | Inference Parameters Overhead Ratio | Training Time Overhead Ratio |
|---|---|---|---|
| ModelNet40 "Rotated" | VN-Pointnet | +0.00 | +0.42 |
| ModelNet40 "Aligned" | VN-Pointnet | +1.25 | +0.42 |
| N-Body Simulation | SEGNN | +0.00 | +0.31 |
| Motion Capture "Run" | EGNO | +0.61 | +0.45 |
| Motion Capture "Walk" | EGNO | +0.39 | +0.45 |
| Conformer Generation | ETFLOW | +0.80 | +0.40 |
| Conformer Generation | DiT-MC | +0.90 | +0.38 |

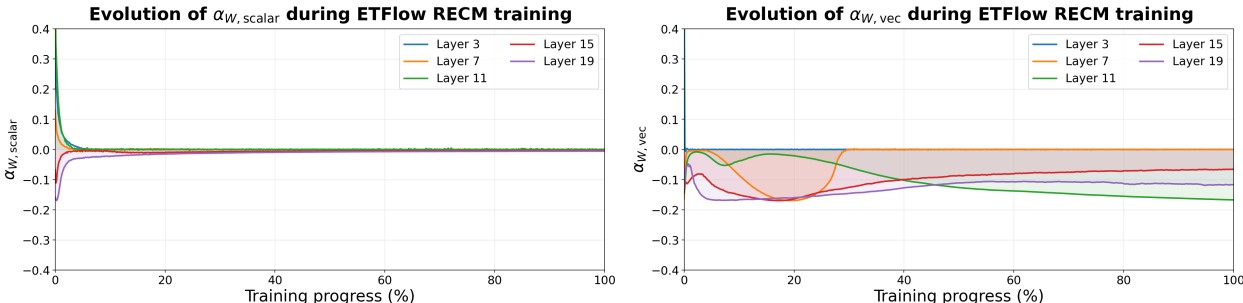

*Figure 6.* Evolution of the constraint modulation parameters of ETFlow+RECM for different layers of the network. (Left) Evolution of $\alpha_{W,\text{scalar}}$ parameter controlling the invariant relaxation of the scalar features, (Right) Evolution of $\alpha_{W,\text{vec}}$ parameter controlling the equivariant relaxation of the vector features.

## C.2. Conformer Generation Metrics

For the task of conformer generation, we evaluate the performance of our generations using the coverage precision and coverage recall metrics.

Given a set of generated conformers $\mathcal{G}$ and a set of reference conformers $\mathcal{R}$, we define the coverage precision as the fraction of conformers that match at least one reference conformer. A generation matches a reference conformer if the generated atom positions have root mean squared distance within a $\delta$ threshold, after we have performed optimal rotational and translational alignment using the Kabsch algorithm (Kabsch, 1976). Similarly, as coverage recall we define the fraction of reference conformers that match at least one generation. For both metrics we use $\delta = 0.75$Å as the threshold.

## C.3. Learned Relaxation in the Conformer Generation Task

As discussed in Section 5.3, the level of relaxation required for the task of conformer generation is actively debated by the community. To gain a better insight into the relaxation level recovered through our proposed framework we inspect the relaxation modulation parameters recovered after training the ETFlow+RECM model, shown in Figure 6. The ETFlow architecture handles both scalar features that are invariant to roto-translations and vector features that are equivariant to roto-translations. For the relaxation of the scalar features, we use an additive term that consists of an unconstrained linear layer acting on all the input features (both vectors and scalars) modulated by $\alpha_{W,\text{scalar}}$. Similarly, for the relaxation of the equivariant features, we use an unconstrained linear layer acting on all the features modulated by $\alpha_{W,\text{vec}}$. We observe that $\alpha_{W,\text{scalar}}$, controlling the unconstrained additive term to scalar (invariant) features, converges to 0 across all layers. In contrast, $\alpha_{W,\text{vec}}$, controlling the unconstrained term on vector (equivariant) features, converges to 0 in early layers but to non-zero values in deeper layers. This suggests that the model benefits from equivariant and invariant layers earlier in the network, and the addition of an orientation bias is necessary only in the later layers' vector features, where the model is required to output a velocity that moves the input sample towards the target distribution.

