# OpenReview forum: "Recurrent Equivariant Constraint Modulation: Learning Per-Layer Symmetry Relaxation from Data"
_ICML.cc/2026/Conference — ICML 2026 spotlight_

### Official Review · Reviewer_KWfK · 2026-02-26

**Soundness:** 4
**Presentation:** 3
**Significance:** 3
**Originality:** 3
**Overall Recommendation:** 5
**Confidence:** 4

**Summary:**

This paper proposes a novel framework called Recurrent Equivariant Constraint Modulation (RECM) to solve the optimization problem in strictly equivariant networks and learn the relaxation level of each layer.
To be specific, the relaxation coefficients of each layer are obtained by an optimization state variable, which is recurrently updated during training using an exponential weighted average and a learnable update rule.
Theoretical analysis demonstrates that RECM converges to recover full equivariance for perfectly symmetric data distributions while retaining the flexibility to learn non-symmetric solutions.
Empirical results show that RECM has outperformed prior relaxation methods in various exact and approximate equivariant tasks.

**Compliance With Llm Reviewing Policy:**

Affirmed.

**Final Justification:**

The rebuttal has addressed my concerns, and I intend to increase the score to 5.

**Key Questions For Authors:**

1. Could the author compare the number of parameters of each relaxed model in the experiments?

2. Could the author explain why they model multiple unconstraint branch $f^{un_i}_{\theta_i}$ in one layer? Also, is the number of unconstraint branch controlled in experiments when comparing other RPP-based relaxed models?

3. What's the learning rate schedule of RECM used in experiments?

4. Would $l_{\theta_t}$ converge to $l_{\theta*}$ that satisfy $E_{(z, y) \sim p}\left[l_{\theta^*}(z, y)\right] \geq C^{\prime} W_1\left(p, p_{C_G}\right)$ when data is not fully symmetric?

5. Given that $\forall \theta_t, l_{\theta_t}$ is bounded by the W1 distance of $p$ and $p_G$, for a fully symmetric setting, it seems that we do not need to update $l_{\theta_t}$ during training. Could the author do a comparison of RECM with fixed $l_{\theta}$ and learned $l_{\theta}$ during training? What are the advantages of learning $l_{\theta}$?

**Limitations:**

yes

**Strengths And Weaknesses:**

## Strengths

1. While there are some optimization-based methods to learn relax equivariant network, RECM consider explicitly model and utilizes the input-target distribution during the optimization process, which is very novel.

2. Sufficient theoretical analysis is provided to make this work more solid.

3. The paper is well-written and well-organized.

## Weaknesses

1. Since RECM introduces a learnable function $l_\theta$ for each layer that would increase the model size and computation complexity, it is natural to rethink the necessity to relax all equivariant layers. To strengthen the evaluation, the authors could include an ablation study comparing the relaxation of specific layer subsets, such as only the final layer versus intermediate layers.

2. Although the evolution of $\alpha_w$ is reported, for data that hold approximate symmetry, such coefficients cannot quantitatively tell the learned task-dependent symmetry. To further compare the models' ability to learn approximate symmetry from the task, the model's equivariance error (EE) should be reported. Authors can carry out experiments on synthetic data with known target equivariance error, and report the discrepancy of the learned model's EE with target EE to demonstrate the model's ability of learning approximate symmetry. The baseline can be RPP and ACE.

3. Since the update of the optimization state variable $h_t$ relies on the input-target distribution of the current batch, the variance of this estimation is naturally tied to the batch size. Including an ablation study on batch size would provide valuable insights into the stability and scalability of this framework.

---

> ### Author Rebuttal · Authors · 2026-03-31
>
> We thank the reviewer for the valuable feedback and suggestions. We would like to address some of the concerns and question raised in the review:
>
> **Regarding relaxing all the equivariant layers versus only a subset of them (W1):**
>
> RECM aims to recover the necessary relaxation levels per layer, using the training data and the task loss without per-layer tuning. Although all layers are relaxed during training, at inference only a subset of them converge to non-zero $\alpha$ (as shown in Figure 2(b) and Figure 4 in the appendix) and act as non-equivariant layers. While only restricting a subset of layers for relaxation during training is possible, it introduces an additional level of hyperparameter search that RECM aims to avoid.
>
> ---
>
> **Toy experiment on Equivariant Error (W2):**
>
> Following the reviewer's suggestion, we trained a model to classify whether a 2D point lies between two ellipses with diameters $(a_1,b_1),(a_2,b_2)$. When $a_1<b_1$, $a_2<b_2$, the task is non-invariant with a known equivariant error (EE), while when $a_1=b_1$, $a_2=b_2$, the task becomes invariant. We use a Vector Neurons MLP as the base model and show how RECM better approximates the ground-truth (GT) EE.
>
> ||Inv. Task|Non-Inv. Task|
> |-|-:|-:|
> |GT-EE|0.0000|0.1162|
> |RPP-EE|0.0097|0.0849|
> |ACE-EE|0.0026| 0.0961|
> |RECM-EE|0.0015|0.0978|
>
> ---
>
> **Relation between the variance of the optimization state and the batch size (W3):**
>
> We will add results showing how the batch size affects the variance of $a_w$ during training, on the ModelNet40 rotated dataset. For each layer, we compute the standard deviation of $a_w$ within a sliding window, then we report the average standard deviation across all windows and all layers. We train models at batch sizes 24$s$ for different scale factors $s$, with learning rate scaled by $s$, and we compute the standard deviation over proportional scaled windows of size 40$/s$.
>
> |Batch-size|12|18|24|30|36|
> |-|-:|-:|-:|-:|-:|
> |$a_w$-std|4.25|4.47|3.92|4.01|3.68|
>
> We can observe that the overall variance remains stable, given the appropriate scale adjustments.
>
> ---
>
> **Comparison of the parameters of each relaxed model (Q1):**
>
> Table 4 in the appendix provides the relative parameter increase for each of our models trained with RECM, relative to the equivariant baseline.
>
> With respect to the baselines, the scheduling-based ES always projects back to the equivariant solution at inference, resulting in no parameter increase. ACE-exact also projects back, while ACE-approx relaxes all layers without proposing any explicit pruning, resulting in a strictly larger number of parameters than RECM.
>
> ---
>
> **Regarding the use of multiple unconstrained terms in one layer (Q2):**
>
> While the benefits of the addition of unconstrained terms during the training of equivariant networks are documented in [1,2], there is limited investigation on which terms help more, given a model architecture. We thus designed a framework agnostic to the specific architectural structure and the optimal way to break equivariance for each layer. We model multiple candidate terms for breaking the symmetry constraint and allow the model to learn to use the appropriate ones. When comparing with the RPP-based baselines, we control for the number of terms by applying the same unconstrained terms
>
> ---
>
> **Learning rate scheduler used (Q3):**
>
> In all experiments, we used the same schedulers as those used by the corresponding baselines models:
> - Point Cloud Classification: Scale lr by 0.7 every 20 steps
> - N-body simulation: No Scheduler
> - Motion Capture: No Scheduler
> - Conformer Generation: Cosine Annealing
>
> ---
>
> **Regarding the convergence of $l_{\theta_t}$ to $l_{\theta^*}$ (Q4):**
>
> Theorem 4.2 asserts that $l_{\theta}(z,y)$ is expressive enough to learn a function whose expectation does not converge to zero in a non-invariant distribution. Although there are no theoretical guarantees of convergence to $\theta^*$, since the optimization uses standard gradient descent, in the experiments on non-invariant distributions, we observed that RECM is able to correctly converge to non-invariant solutions (Figure 2(b))
>
> ---
>
> **Regarding the need for learning $l_{\theta_t}$ (Q5):**
>
> Updating the parameters $\theta_t$ allows the network, even in fully symmetric tasks, to control the convergence rate of each individual layer in a manner that aligns with the task loss. If we remove this learning, the convergence of each layer will depend solely on the arbitrary initialization of $l_\theta$, with no control or supervision from the training task. Furthermore, for not fully symmetric tasks, learning $\theta_t$ allows the model to learn update rules that align with the task loss, and do not converge to equivariant solutions.
>
> [1] Manolache et al.  "Learning (Approximately) Equivariant Networks via Constrained Optimization" (NeurIPS 2025)
>
> [2] Pertigkiozoglou et al. "Improving Equivariant Model Training via Constraint Relaxation" (NeurIPS 2024)

---

> > ### Author Rebuttal · Reviewer_KWfK · 2026-04-03
> >
> > Thanks for the rebuttal. My concerns are fully solved, and I am happy to raise the score.

---

> > > ### Author Response · Authors · 2026-04-08
> > >
> > > We thank the reviewer for the constructive feedback and questions about our work, and for providing a positive recommendation for our submission.

---

### Official Review · Reviewer_QrKQ · 2026-03-06

**Soundness:** 3
**Presentation:** 3
**Significance:** 3
**Originality:** 3
**Overall Recommendation:** 4
**Confidence:** 3

**Summary:**

They propose RECM, a method that learns how much equivariance each layer should keep or relax during training. Their model mixes equivariant and non-equivariant components and updates the relaxation level recurrently based on training signal. The paper's theory argues learned relaxation values are controlled by a symmetry gap quantity and experiments cover exact and approx symmetry settings including molecular conformer generation.

**Compliance With Llm Reviewing Policy:**

Affirmed.

**Final Justification:**

The rebuttal resolved my concerns clarifying RECM’s tuning is tied more to architecture than task, showing added overhead is modest for large models, giving clearer picture of where the method helps on GEOM-Drugs, so I remain positive. will keep original score.

**Key Questions For Authors:**

1. How sensitive is RECM to number/type of non-equivariant candidates and to the choice of state dim m?

2. What is practical compute/memory overhead relative to ACE or schedule baselines?

3. How brittle is the pruning threshold for small alpha at inference?

4. Is there more intuition for what recurent update is learning in harder approx symmetry tasks?

5. On GEOM-Drugs, where do you think RECM helps the most and where does it not help much?

**Limitations:**

No. I want them to be more upfront about strength of assumptions used in their convergence and also about the fact that their method still introduces several design knobs of its own!

**Strengths And Weaknesses:**

In practice, the amount of relaxation that works best is annoyingly task dependent, so I really like the paper’s idea that you can learn the right amount of equivariance from data instead of setting it by hand layer by layer. Their theory is more substantial than I expected and empirical section is broad enough to be plausible.

However, I'm not sure how automatic the method really is! RECM still brings along a few design choices, like candidate non-equivariant terms, state dim, update dynamics, thresholds, and so on. So it removes one kind of tuning but not all tuning! Theory has reasonable assumptions but are hard to verify in actual deep learning training. Gains are good but not massive in every setup, which is fine, though the paper sometimes sounds a bit more 'works here and there' than their results justify.

---

> ### Author Rebuttal · Authors · 2026-03-31
>
> We thank the reviewer for recognizing the significance of our proposed framework and for providing valuable feedback. We would like to address the questions and concerns raised in the review:
>
> **Regarding required hyperparameter tuning (W1):**
>
> While our method still has design choices that affect performance, they control only the expressivity of the update rule and its convergence, without introducing an implicit bias towards the desired level of relaxation. As a result, while hyperparameter tuning is still necessary for a given model, we can use the same hyperparameters to recover different levels of relaxation when training the same model on different data distributions, as shown in the point cloud classification experiment in Section 5.1. In contrast, prior methods like ACE require tuning for each individual training distribution. We thank the reviewer for the remark, and we will add a clearer discussion on our exact contribution and what type of tuning our method aims to remove.
>
> ---
>
> **Sensitivity to the choice of state dim m and the number/type of non-equivariant candidates (Q1)**
>
> To compute the dimension $m$, we performed a grid search on ModelNet40. The resulting instance accuracies are shown below:
>
> |m|4|8|16|32|
> |-|-|-|-|-|
> |VN-PointNet RECM|0.78|0.80|0.80|0.79|
>
> We can observe that the performance of RECM is relatively robust around $m=16$, while $m=32$ provides no additional benefit. Notably, $m$ was tuned only on the ModelNet40 dataset and was able to generalize and provide performance improvement in the subsequent experiments without requiring any additional tuning.
>
> Regarding sensitivity to the number of non-equivariant terms, we observed that the linear layer and bias terms contribute the most to the performance gains, with the additive noise only providing a +2\% increase in accuracy in the point cloud classification dataset.
> While additive noise does not seem essential, especially for regression, our method is robust to its inclusion.
>
> ---
>
> **Compute/Memory overhead relative to ACE or schedule baselines (Q2)**
>
> The overhead of RECM compared to the ACE and schedule baseline comes only from the additional forward passes of the update rule $l_\theta$, which occur only during training. Since each $l_\theta$ requires forward passes of a small MLP with $n$  input and $m=16$ hidden dimensions, the computational and memory overhead is in the order of $O(nm)$. Given that the computational and memory cost of the typical linear layer, assuming equal input and output dimensions, is in the order of $O(n^2)$, the proportional overhead decreases with network size. This overhead ranges from 9ms overhead in a 23ms iteration (40\% overhead) in the smaller VN-PointNet network, to 17ms overhead in a 512ms iteration (3\% overhead) in the ETFlow network.
>
> ---
>
> **Sensitivity to the pruning threshold (Q3)**
>
> The $\alpha$ threshold is only relevant at inference to remove near-zero-contribution parameters. The choice depends on the desired tradeoff between task performance and compute/memory efficiency. In all of our tasks, we observed that a threshold of 0.01 is sufficient to distinguish between parameters that have converged to 0 and those actively contributing to the loss, and that the model's performance is robust to its value. We observed significant changes (more than 1\%) in the model performance and training loss when we started pruning parameters above the 0.02 threshold. Thus, the choice of 0.01 is a conservative choice with sufficient margin.
>
> ---
>
> **Intuition on what the recurrent update is learning in harder approximate symmetry tasks, and where it helps most in the GEOM-Drugs dataset (Q4-5)**
>
> We thank the reviewer for these questions. In order to provide a better intuition on what the recurrent update is learning in harder approximate symmetry tasks, we will add a figure showing the evolution of the $\alpha$ parameters in the ETFlow model trained on the GEOM-Drugs.
>
> We describe some of our observations: The ETFlow model simultaneously learns both invariant scalar features and equivariant vector features given an input conformer sample. We observe that the $\alpha$ parameter controlling the symmetry constraint on the scalar features converges to 0, while for vector features, only deeper-layer's $\alpha$ parameters converge to non-zero values. This suggests that the model benefits from equivariant and invariant layers earlier in the network, and the addition of an orientation bias is necessary only in the later layers, where the model is required to output a velocity that moves the input sample towards the target distribution. This observation confirms the results of previous work [1], where symmetry-constrained layers are beneficial in the earlier stages of the models, while symmetry relaxation is helpful in the latter layers to match the symmetries of the output distribution.
>
> [1] M. Weiler and G. Cesa "General E(2)-Equivariant Steerable CNNs" (NeurIPS 2019)

---

> > ### Author Rebuttal · Reviewer_QrKQ · 2026-04-04
> >
> > Thanks! Clarifying RECM's hyperparameters are tied to the architecture (not the task!) is a major practical selling point which you should emphasize this in the revision. Compute overhead dropping to just 3% for larger models and the GEOM-Drugs clarification fully resolves my concerns. I will keep my positive score.

---

> > > ### Author Response · Authors · 2026-04-08
> > >
> > > We would like to thank the reviewer for the valuable feedback and for the positive assessment of our work. We also appreciate the reviewer’s clarifying questions. In the updated manuscript, we will emphasize both the connection between the RECM's hyperparameters and the model architecture (rather than the task), as well as how the RECM's compute overhead scales relative to the baseline methods.

---

### Official Review · Reviewer_J6is · 2026-03-13

**Soundness:** 2
**Presentation:** 2
**Significance:** 1
**Originality:** 2
**Overall Recommendation:** 3
**Confidence:** 4

**Summary:**

This paper addresses the limitation that strict equivariance constraints may not be necessary at every layer of a network. To tackle this, the authors propose a method that adaptively adjusts the strength of these constraints during the optimization process using a recurrent module.

**Compliance With Llm Reviewing Policy:**

Affirmed.

**Final Justification:**

The rebuttal discussed my concerns well, though I recognize that expanding the paper's coverage to include these various perspectives is challenging given the current constraints. Since my remaining feedback may be subjective compared to the other reviewers' view, I will keep my score as-is, but I am happy to support AC's decision.

**Key Questions For Authors:**

.

**Limitations:**

Yes

**Strengths And Weaknesses:**

## Strength
### 1. Intuitive and Highly Relevant Method
The paper proposes an intuitive and much-needed approach to address the rigidity of current equivariant models.
### 2. Significant Theoretical Motivation
It raises a critical question regarding the absolute correctness of equivariant and invariant inductive biases, successfully offering a practical solution to relax them.

## Weakness
### 1. Sensitivity of Optimization and Generalization (Soundness)
While adaptively balancing invariance and equivariance is a practical goal, the optimization path is highly sensitive to this balance, which in turn depends heavily on the sparsity and complexity of the data. Even if the theoretical convergence point aligns with a strict equivariance constraint, the adaptive regularization of this inductive bias can lead to highly variable empirical outcomes. The authors must explicitly discuss the generalization capabilities of their adaptive strategy. If the adaptation cannot generalize broadly, the paper should clearly define the specific conditions or data environments where this method is guaranteed to succeed.
### 2. Outdated and Limited Baselines (Significance)
The reliance on VN-PointNet and ModelNet requires stronger justification. While acceptable for a toy analysis, these are far from the state-of-the-art architectures used in practice, which severely limits the demonstrated generality of the method. Furthermore, baselines like Tensor Field Networks (TFN) and Equivariant Filter (EF) are largely outdated.
### 3. Narrow Architectural Scope (Significance)
The paper focuses on explicit constraint handling, but many modern models handle equivariance balancing implicitly. While the authors include EGNN and EGNO, these are exclusively Graph Neural Networks. To demonstrate true broad impact, the authors need to compare their approach across a wider variety of base architectures beyond just GNNs.
### 4. Lack of Justification in Comparison with Complex/Unknown Symmetries Learning (Significance)
The current symmetry research community is rapidly moving toward detecting and learning unknown, unrestricted, and noisy symmetries in complex environments. Testing solely on simple, known symmetries (like basic rotation or translation) limits the paper's relevance. The authors need to discuss how their symmetry constraint balancing applies to uncertain, complex symmetries in practice, and compare it against other contemporary methods tackling soft symmetry learning.

---

> ### Author Rebuttal · Authors · 2026-03-31
>
> We thank the reviewer for the valuable feedback. We would like to address some of the concerns raised in the review:
>
> **Regarding the generalization capabilities of RECM (W1):**
>
> We believe that our work makes sufficient effort to showcase the generalization of the proposed framework. First we provided theoretical results that showcase the explicit assumptions required for the method to converge to the required level of relaxation. These assumptions include the compactness of the group of interest along with the convergence of the latent distribution, which, as we discussed, can be achieved by a commonly used learning rate scheduler. Empirically, we validate the generalization of our method by evaluating on a broad set of tasks, including point cloud classification, motion capture prediction, and molecular conformer generation, that correspond to training datasets of different sizes and tasks with different degrees of required symmetry. We would appreciate if the reviewer has additional suggestions on what aspect we can improve in the presentation of our framework.
>
> ---
>
> **Regarding the outdated and limited baselines (W2):**
>
> In our paper, the ModelNet classification task and the VN-PointNet model are used as a simple base task and model that allows us to run large-scale ablation studies without a prohibitive amount of computational resources. After using this simple task for tuning, we showcase how it generalizes in larger scale experiments.
>
> In the motion capture experiments, the main baseline used for comparison is the EGNO, which is currently one of the state-of-the-art models relevant to this problem, and the TFN and EF are only provided as a context for the performance of previous methods.
>
> Additionally, regarding the use of state-of-the art models: since the tasks of point cloud classification, N-body simulation and motion capture prediction have been extensively studied, with the already existing models achieving optimal performance, we focus on the more challenging task of conformer generation, which is currently an active research topic as shown in [1,2,3]. In this latter large-scale and challenging task, we utilize the current state-of-the-art baseline models, ETFlow, DiTMC, to showcase how our method is able to improve their performance.
>
> ---
>
> **Regarding the narrow architecture scope concern (W3):**
>
> In the experimental evaluation of our method, we use a broader range of network architectures that go beyond GNNs. Namely, in Section 5.1, a simple VN-PointNet architecture is used, while in Section 5.3, we use a flow matching and a diffusion transformer (ETFlow and DiTMC).
>
> ---
>
> **Regarding the lack of comparison with complex/unknown symmetry learning (W4)**
>
> The focus of our paper is to design an optimization algorithm that actively adapts the level of relaxation of a hypothesized symmetry, which is a research problem that is currently actively investigated by the community, as shown in Pertigkiozoglou et al. (NeurIPS 2024) and Manolache et al. (NeurIPS 2025). Additionally, in Section 5.3, we showcase how our method is relevant to discovering the level of symmetry for the challenging conformer generation task, where the community actively debates the benefits of rotational equivariant networks. While discovering unknown complex symmetries is an interesting and significant research question, it is out of the focus of the current work.
>
>
> [1] Jing et al. "Torsional diffusion for molecular conformer generation" (NeurIPS 2022)
>
> [2] Wang et al. "Swallowing the bitter pill: Simplified scalable conformer generation." (ICML 2024)
>
> [3] Frank et al. "Sampling 3d molecular conformers with diffusion transformers." (NeurIPS 2025)

---

> > ### Author Rebuttal · Reviewer_J6is · 2026-04-03
> >
> > Thank you to the authors for their efforts with the rebuttal. I have carefully reviewed the responses and can confirm that some of my issues have been resolved.
> >
> > However, while the explanations clarified the boundaries of the research, which is appreciated and I still believe this research topic is necessary for the community, they do not sufficiently resolve the underlying weaknesses of the current version. Therefore, I will maintain my current score.

---

> > > ### Author Response · Authors · 2026-04-08
> > >
> > > We thank the reviewer for the feedback and for recognizing the relevance of the specific research topic to the community. We are pleased that the explanations provided in the rebuttal helped clarify the specific aims of our work, and we will incorporate them in the updated version of the paper.

---

### Official Review · Reviewer_hHSP · 2026-03-18

**Soundness:** 3
**Presentation:** 3
**Significance:** 4
**Originality:** 2
**Overall Recommendation:** 5
**Confidence:** 3

**Summary:**

The paper proposes a new method for per-layer symmetry learning. The paper proposes to relax symmetry constraints in the architecture and make them learnable from data, removing the need for manually tuned schedules or predefined relaxation levels. To do so, the authors introduce Recurrent Equivariant Constraint Modulation (RECM).

**Compliance With Llm Reviewing Policy:**

Affirmed.

**Final Justification:**

I thank the authors for their detailed and thorough rebuttal. Most of the issues I raised have been adequately addressed through clarifications, discussion, and provided additional quantitative experiments. Overall, the response has strengthened the paper and clarified its contributions. I have since raised my score and recommend this paper for acceptance.

**Key Questions For Authors:**

(Q.1): The proofs seem tailored to compact groups. Is this an issue for non-compact groups, like SE(3) which are considered in the experiments?
(Q.2): When do we expect the theory or objective not to hold? Or do we always expect them to work well.


Related to the last-mentioned weakness, namely the missing comparisons, or even mention, of existing approaches:

On the parameterization side, [1] is mentioned, and the paper follows it by parameterizing relaxed equivariance through a residual pathway: a sum of a non‑equivariant (linear) layer and an equivariant layer, which spans the equivariant subspace, with coefficients controlling the amount of equivariance. Similar to other approaches, and as recognized by the authors in Sec. 2, competing methods often use equivariance‑promoting regularizers to learn the degree of equivariance and to avoid collapse into trivial non‑equivariant solutions. (Q.3:) Why are such objectives/update rules not compared against as baseline?

Further, regarding the objective, other methods with similar parameterizations have considered (sometimes probabilistically principled) objectives to infer the amount of symmetry, such as based on differentiating validation data [2], or the approach by [3] that uses Bayesian model selection and similar to this work claims learning "per-layer" symmetries from data. (Q.4:) Are the authors aware of this line of work, or is there a reason why it was not deemed relevant to compare against? Since such methods exist, the claim that “(other works) lack a principled way to choose [where] to relax equivariance constraints and by [how much]” seems too strong. The paper should recognize these approaches and ideally compare against them or explain why such a comparison is not appropriate.

[1] Finzi, Marc, Gregory Benton, and Andrew G. Wilson. "Residual pathway priors for soft equivariance constraints." Advances in Neural Information Processing Systems 34 (2021): 30037-30049.
[2] Zhou, Allan, Tom Knowles, and Chelsea Finn. "Meta-learning symmetries by reparameterization." arXiv preprint arXiv:2007.02933 (2020).
[3] van der Ouderaa, Tycho, Alexander Immer, and Mark van der Wilk. "Learning layer-wise equivariances automatically using gradients." Advances in Neural Information Processing Systems 36 (2023): 28365-28377.

**Limitations:**

-

**Strengths And Weaknesses:**

Overall, I regard this paper as a very well-written and strong contribution. The work is novel and interesting and deals with important line of research.

Strengths:
- Symmetry constraints form one of the most important inductive biases that can be incorporated in machine learning / deep learning architectures. However, constraints can be too strict or the amount of relaxation can be hard to set manually. This paper offers an approach to achieve this, which is a relevant direction from a research perspective.
- The update rule proposed in the paper is novel and an interesting solution to providing a training signal for learning per-layer symmetry constraints.
- The paper does not only propose a method, but also offers theoretical guarantees which substantiate the method making the method more grounded.
- Beyond theoretical and methodological proposals, the method empirically presents improved performance, as on the GEOM‑Drugs dataset, demonstrating consistent gains and practical applicability.

Weaknesses:
- The main optimization state update formula in Eq. 1 seems like a very interesting and novel contribution. However, I think it would strengthen the paper if the main text could more extensively discuss the motivation or workings of the update rule, also in relation to other objectives in literature.
- Would be interesting to obtain more discussion on when RECM fails or converges slowly (e.g., high‑noise approximate symmetries). Or do we always expect it to work? How does this compare to simpler regularization-based approaches or more sophisticated Bayesian approaches.
- Other methods for symmetry learning have been proposed, both on the parameterization (how symmetry constraints are relaxed) and on the objective/update rule side. These works are not compared against and sometimes not mentioned at all. (see questions)

---

> ### Author Rebuttal · Authors · 2026-03-31
>
> We thank the reviewer for recognizing both the theoretical and empirical contributions of our work and for the constructive feedback and suggestions. We would like to address the questions raised in the review:
>
> **Regarding the motivation of the update formula and its comparison with previous approaches**
>
> We appreciate the suggestion of the reviewer, and we will add a more detailed description of the motivation behind the update formula in Eq. 1 and its possible limitations. Previous works [3,4] optimized relaxation parameters by introducing alternative objectives and performing gradient descent. In [3], such an objective is derived from Bayesian model selection, while in [4] it is derived from the dual of a constrained optimization problem. Such objectives can introduce implicit biases competing against the task loss.  In our work, we investigated how a learnable update rule can guarantee convergence properties by constraining the learnable function's form, allowing us to use only the task objective to learn both model parameters and the update rule.
>
> While removing implicit biases provides additional flexibility, learning the update rule during training, compared to a simple gradient step on a predefined objective, can slow down the convergence to a stable solution when distributions are highly non-invariant.
>
> ---
>
> **Regarding Questions Q1 and Q2**
>
> As noted by the reviewer, the current framework is tailored to compact groups because it requires a Haar measure and input-target distributions that are invariant under all group actions. The SE(3) models we used achieve translation equivariance by expressing relative positions as differences of locations in $\mathbb R^3$. In our experiments, we keep these relative descriptions, and we only relax the weight constraints that control the rotational equivariance. In other words, we only relax equivariance under SO(3) while preserving translational equivariance.
>
> If we apply the same framework to a non-compact group, convergence guarantees do not apply, leaving the update rule unconstrained to converge to non-equivariant solutions even for fully symmetric distributions. We will add a detailed remark regarding the compactness requirement in the updated manuscript.
>
> ---
>
> **Regarding Questions Q3 and Q4**
>
>  We thank the reviewer for the suggested related work. We will add a detailed discussion on their connections to our work:
>
> Our framework follows closely the parametrization proposed in [1]. The submitted manuscript already includes a comparison in point cloud classification. Additionally, we will include comparisons in the experiments of N-body simulation and Motion Capture Prediction.
>
> In [2], the authors take an approach different from our framework: they assume a fixed set of degrees of freedom/parameters and learn the appropriate weight-sharing constraints among them in a meta-learning fashion while training on multiple tasks. As a result, they do not adaptively control the relaxation level of a predefined symmetry constraint, but instead they learn the appropriate weight-sharing constraint and group structure that is common within multiple tasks.
>
> In [3] (ELLA), the authors utilize the RPP constraint relaxation, similar to ACE and ours, but propose to optimize the parameters $\eta$ controlling the level of relaxation through a Bayesian model selection framework. A main limitation of [3] is the need to compute the marginal likelihood of the data conditioned on the $\eta$ parameters, which requires the computation of the model's Hessian. Even when using specialized KFAC approximation techniques,  the approximation of the Hessian limits the scalability of the method to large or specialized models. Additionally, the simultaneous optimization of the model's parameters and the relaxation level via maximization of the marginal likelihood introduces complex optimization dynamics without any convergence guarantees.
>
> We added comparisons both with RPP and ELLA in the tasks of point cloud classification, N-body simulation, and motion capture prediction. In the tables below, RPP uses the constrained relaxation parametrization optimized with gradient descent, while ACE and ELLA utilize additional objectives/regularizers to better control relaxation convergence.
>
> - **Point Cloud Classification**
> | |Rotated Inst|Rotated Cls|Aligned Inst|Aligned Cls|
> |-|:-:|:-:|:-:|:-:|
> |VN-PointNet RPP|0.77|0.71|0.89|0.86|
> |VN-PointNet ELLA |0.78|0.71|0.90|0.86|
> |VN-PointNet RECM |0.80|0.74|0.90|0.86|
>
> - **N-body Simulation**
> | |MSE|
> |-|:-:|
> |SEGNN RPP|4.1±0.17|
> |SEGNN ELLA|3.9 ± 0.11|
> |SEGNN ACE-Exact|3.8 ± 0.16|
> |SEGNN RECM|3.7±0.22|
>
>
> - **Motion Capture Prediction**
> ||MSE (RUN)|MSE (WALK)|
> |-|:-:|:-:|
> |EGNO-RPP|28.0±1.6|7.2±0.2|
> |EGNO-ELLA| 29.9 ± 1.1| 7.0 ± 0.4|
> |EGNO-ACE APPR|23.8 ±1.5| 7.4 ± 0.2|
> |EGNO-RECM|22.6±0.8|6.6 ± 0.5|
>
> [4] Manolache et al.  "Learning (Approximately) Equivariant Networks via Constrained Optimization" (NeurIPS 2025)

---

> > ### Author Rebuttal · Reviewer_hHSP · 2026-04-07
> >
> > I thank the authors for their detailed and thorough rebuttal. Most of the issues I raised have been adequately addressed through clarifications, discussion, and provided additional quantitative experiments. Overall, the response has strengthened the paper and clarified its contributions. I have since raised my score and recommend this paper for acceptance.

---

> > > ### Author Response · Authors · 2026-04-08
> > >
> > > We would like to thank the reviewer for the constructive feedback and suggestions that helped us strengthen the presentation of our work. We appreciate the positive assessment of our submission.

---

### Decision · Program_Chairs · 2026-04-30

**Decision:**

Accept (spotlight)

**Comment:**

This paper proposes RECM, a mechanism for adaptively learning per-layer symmetry relaxation levels in equivariant neural networks. The method comes with theoretical convergence guarantees and is validated across point cloud classification, N-body simulation, motion capture prediction, and molecular conformer generation. The paper addresses an important problem, proposes a novel and theoretically grounded solution, and demonstrates consistent empirical gains. I recommend acceptance.

Three of four reviewers recommended acceptance (5, 5, 4), with all three indicating their concerns were fully resolved by the rebuttal.
Reviewer J6is (score 3, confidence 4) expressed concerns about baseline breadth, architectural scope, and the lack of engagement with unknown symmetry discovery.  I have read the authors' rebuttals. The additional experiments provided during rebuttal (comparisons with RPP/ELLA, equivariance error analysis, sensitivity studies) meaningfully strengthened the paper. I encourage the authors to incorporate these additions and a more explicit discussion of limitations (compactness assumption, convergence speed for highly non-invariant distributions) into the camera-ready version.